

**Rapid transition in winter aerosol composition in Beijing from 2014 to**
**2017: response to clean air actions**
Haiyan Li[1,a], Jing Cheng[2], Qiang Zhang[2], Bo Zheng[1], Yuxuan Zhang[2], Guangjie Zheng[1], Kebin He[1,3]
[1] State Key Joint Laboratory of Environment Simulation and Pollution Control, School of Environment, Tsinghua University,
Beijing 100084, China
[2] Ministry of Education Key Laboratory for Earth System Modeling, Department of Earth System Science, Tsinghua University,
Beijing 100084, China
[3] State Environmental Protection Key Laboratory of Sources and Control of Air Pollution Complex, Tsinghua University, Beijing
100084, China
[a]present address: Institute for Atmospheric and Earth System Research/Physics, Faculty of Science, University of Helsinki, 00014
Helsinki, Finland
*Correspondence:* Qiang Zhang (qiangzhang@tsinghua.edu.cn)
**Abstract**. The clean air actions implemented by the Chinese government in 2013 have led to significantly improved air quality in
Beijing. In this work, we combined the in-situ measurements of the chemical components of submicron particles ($PM_1$) in Beijing
during the winters of 2014 and 2017 and a regional chemical transport model to investigate the impact of clean air actions on
aerosol chemistry and quantify the relative contributions of anthropogenic emissions, meteorological conditions, and regional
transport to the changes in aerosol chemical composition from 2014 to 2017. We found that the average $PM_1$ concentration in
winter in Beijing decreased by 49.5% from 2014 to 2017 (from 66.2 μg m$^{-3}$ to 33.4 μg m$^{-3}$). Sulfate exhibited a much larger decline
than nitrate and ammonium, which led to a rapid transition from sulfate-driven to nitrate-driven aerosol pollution during the
wintertime. Organic aerosol (OA), especially coal combustion OA, and black carbon also showed large decreasing rates, indicating
the effective emission control of coal combustion and biomass burning. The decreased sulfate contribution and increased nitrate
fraction were highly consistent with the much faster emission reductions in sulfur dioxide ($SO_2$) due to phasing out coal in Beijing
compared to reduction in nitrogen oxides emissions estimated by bottom-up inventory. The chemical transport model simulations
with these emission estimates reproduced the relative changes in aerosol composition and suggested that the reduced emissions in
Beijing and its surrounding regions played a dominant role. The variations in meteorological conditions and regional transport
contributed much less to the changes in aerosol concentration and its chemical composition during 2014-2017 compared to the
decreasing emissions. Finally, we observed that changes in precursor emissions also altered the aerosol formation mechanisms.
The decreased $SO_2$ emissions suppressed the rapid formation of secondary sulfate through heterogeneous reactions. The observed
explosive growth of sulfate at a relative humidity (RH) greater than 50% in 2014 was delayed to a higher RH of 70% in 2017.
Thermodynamic simulations showed that the decreased sulfate and nitrate concentrations have lowered the aerosol water content,
particle acidity, and ammonium particle fraction. The results in this study demonstrated the response of aerosol chemistry to the
stringent clean air actions and identified that the anthropogenic emission reductions are a major driver, which could help to further
guide air pollution control strategies in China.
**1 Introduction**
Beijing, the capital of China, is one of the most heavily polluted cities in the world (Lelieveld et al., 2015), and it frequently
experiences severe and persistent haze pollution episodes in winter (Guo et al., 2014). For example, in January 2013, the daily
concentration of ambient particles with an aerodynamic diameter less than 2.5 μm ($PM_{2.5}$) reached a record high of 569 μg m$^{-3}$ in
Beijing (Ferreri et al., 2018), which was over 20 times higher than the World Health Organization standard (25 μg m$^{-3}$ for daily



average PM$_{2.5}$). As a complex mixture of many different components, ambient aerosols have a range of chemical compositions and
originate from various emission sources and formation processes in the atmosphere (Seinfeld and Pandis, 2012). The adverse
effects of aerosols on visibility (Pui et al., 2014), climate (IPCC, 2013), and human health (Pope et al., 2009) are intrinsically
related to the chemical composition of particles.
To tackle severe aerosol pollution, the Chinese State Council implemented the Air Pollution Prevention and Control Action Plan
(denoted as clean air actions) in September 2013, which is the most stringent pollution mitigation policy ever in China. As a
consequence, China's anthropogenic emissions have declined by 59% for SO$_2$, 21% for NO$_x$, 32% for organic carbon (OC), and
28% for black carbon (BC) during 2013-2017 (Zheng et al., 2018). The annual average PM$_{2.5}$ concentration in Beijing decreased
by 35.6% from 2013 to 2017, reaching 58 μg m$^{-3}$ in 2017. Combining the bottom-up emission inventory and chemical transport
model simulations, our recent study (Cheng et al., 2019) quantified the relative contributions of meteorological conditions,
emission reductions from surrounding regions, and emission reductions from local sources to the decrease in PM$_{2.5}$ concentration
in Beijing during 2013-2017. While changes in meteorological conditions partially explained air quality improvement in Beijing
in 2017, local and regional emission controls played major roles. In addition, the aerosol chemical composition is expected to
change correspondingly due to the rapid reductions in precursor emissions, which is not well understood yet because the chemical
components of PM$_{2.5}$ are not measured by China's monitoring network. A few studies have examined the change in aerosol
composition in Beijing after 2013, including a semicontinuous measurement of carbonaceous aerosols during 2013-2018 (Ji et al.,
2019) and an aerosol mass spectrometry study comparing aerosol composition and size distribution between 2014 and 2016 (Xu
et al., 2018). However, neither performed a comprehensive assessment of all the main factors affecting aerosol concentration and
its composition. A deep understanding of how the aerosol composition has changed since the clean air actions were activated and
the possible linkage between them is urgently needed.
The chemical composition of PM$_{2.5}$ is mainly affected by three factors: precursor emissions, meteorological conditions, and
regional transport patterns. Emissions are typically the main driver of aerosol composition changes. During 2005-2012, the sulfate
concentration in China decreased, while the nitrate concentration increased, which was caused by the considerable reduction in
SO$_2$ emissions but limited control of NO$_x$ (Geng et al., 2017). Based on the measurements of organic aerosol (OA) composition in
Beijing, a larger decrease in secondary OA than primary OA was found during the 2014 Asia-Pacific Economic Cooperation
summit due to the strict emission controls (Sun et al., 2016). Meteorological conditions affect aerosol composition by changing
emissions, chemical reactions, and transport and deposition processes (Mu and Liao, 2014). For example, increases in relative
humidity (RH) enhance the secondary formation of sulfate through heterogeneous reactions (Zheng et al., 2015; Cheng et al., 2016),
and decreases in temperature favor particulate nitrate formation by facilitating gas-to-particle partitioning (Pye et al., 2009; Li et
al., 2018). With chemical transport model simulations in China for the years 2004-2012, Mu and Liao (2014) demonstrated that
due to the large variations in meteorological parameters in North China, all aerosol species showed large corresponding interannual
variations. Furthermore, aerosol characteristics in Beijing are influenced by regional transport from adjacent polluted regions.
Polluted air masses from the southern regions contributed more secondary inorganic aerosols (SIAs) than primary aerosols in
Beijing (Zhang et al., 2014; Du et al., 2018).
Following our previous work (Cheng et al., 2019), the main objective of this study is to investigate the impact of clean air actions
on changes in aerosol chemical composition from 2014 to 2017. With both the in-situ observations of aerosol species in Beijing
during the winters of 2014 and 2017 and model simulations for the corresponding periods, this work provides the opportunity for
a detailed evaluation of the underlying drivers. First, changes in aerosol characteristics are illustrated for inorganics and organics
by comparing aerosol measurements in 2014 and 2017. Then, the relative importance of different factors in varying aerosol
composition is assessed by combining direct observations and model simulations, including synoptic conditions, emission changes,





regional transport and formation mechanisms. Last, we show that the transition in aerosol characteristics influenced particle
properties, such as aerosol water content (AWC) and particle acidity, which in turn affects secondary aerosol formation.

## 2 Experimental methods

### 2.1 Ambient sampling and instrumentation

Online aerosol measurements were performed in urban Beijing during the winters of 2014 (from 6 December 2014 to 27 February
2015) and 2017 (from 11 December 2017 to 2 February 2018). The sampling site is located on the roof of a three-story building
on the campus of Tsinghua University (40.0° N, 116.3° E), which is surrounded by school and residential areas. No major industrial
sources are situated nearby. An Aerodyne Aerosol Chemical Speciation Monitor (ACSM) was deployed for the real-time chemical
observations of nonrefractory $PM_1$ (NR-$PM_1$), including organics, sulfate, nitrate, ammonium, and chloride. A detailed description
of the instrument can be found in Ng et al. (2011a). The mass concentration of BC in $PM_1$ was measured using a multiangle
absorption photometer (MAAP, model 5012; Petzold and Schönlinner, 2004). In addition, the total $PM_{2.5}$ mass was simultaneously
recorded with a PM-712 monitor based on the β-ray absorption method (Kimoto Electric Co., Ltd., Japan). For gaseous species,
the mixing ratios of $SO_2$, $NO_x$, CO, and $O_3$ were monitored by a suite of commercial gas analyzers (Thermo Scientific). The
meteorological parameters, including temperature, RH, wind speed (WS), and wind direction (WD), were obtained from an
automatic meteorological observation instrument (MILOS520, VAISALA Inc., Finland).

### 2.2 ACSM data analysis

The ACSM data were analyzed using the standard analysis software within Igor Pro (WaveMetrics, Inc., Oregon USA). Default
relative ionization efficiencies (RIEs) were applied to organics (1.4), nitrate (1.1), and chloride (1.3), while the RIEs of ammonium
and sulfate were experimentally determined through calibrations with pure ammonium nitrate and ammonium sulfate, respectively.
A composition-dependent collection efficiency (CE) algorithm was used to account for the incomplete detection of aerosol particles
(Middlebrook et al., 2012). As shown in Fig. S1, the total measured $PM_1$ mass (NR-$PM_1$ plus BC) correlated well with the $PM_{2.5}$
obtained from PM-712 ($r^2$ = 0.80 and 0.87 for 2014 and 2017, respectively). On average, $PM_1$ accounted for 68% and 80% of the
total $PM_{2.5}$ in Beijing during the winters of 2014 and 2017.
The ACSM provides unit-mass-resolution mass spectra of submicron particles, facilitating source apportionment via factor analysis.
In this study, positive matrix factorization (PMF) was implemented to resolve OA into various sources using a multilinear engine
(ME-2; Paatero, 1999) via the SoFi toolkit (Source Finder; Canonaco et al., 2013). The *a* value approach allows for the introduction
of a priori factor profile or time series to reduce the rotational ambiguity and obtain a unique solution. The spectra and error
matrices of organics were pretreated based on the procedures given by Ulbrich et al. (2009) and Zhang et al. (2011). Ions up to *m/z*
were considered in this study given the interferences of the internal standard of naphthalene at *m/z* 127-129 and the low signal-
to-noise ratio of larger ions. For the winter of 2014, a reference hydrocarbon-like OA (HOA) profile from Ng et al. (2011b) was
introduced into the ME-2 analysis, varying *a* value from 0 to 1. After a detailed evaluation of the factor profiles, time series, diurnal
variations, and correlations with external tracers, an optimal solution with four factors was finally accepted, with an *a* value of 0.
Figure S2 shows the source apportionment results with three primary factors, i.e., HOA, coal combustion OA (CCOA), and biomass
burning OA (BBOA), and one secondary factor, oxygenated OA (OOA). For the 2017 dataset, the mass spectral profiles of HOA,
CCOA, and BBOA from the ME-2 analysis for 2014° were adopted to constrain the model performance. Similarly, a four-factor
solution with HOA, BBOA, CCOA, and OOA was selected for the winter of 2017, which allowed a better comparison of the OA
sources between 2014 and 2017.



## 2.3 WRF-CMAQ model

The Weather Research and Forecasting (WRF) model, version 3.8, and the Community Multiscale Air Quality (CMAQ) model, version 5.1, were applied to evaluate the impact of meteorological changes, regional transport and emission variations on the $PM_{2.5}$ concentration in Beijing in winter. The simulated area was designed as three nested domains, and the innermost area covered Beijing and its surrounding regions (including Tianjin, Hebei, Shanxi, Henan, Shandong and Inner Mongolia), with a horizontal resolution of 4 km × 4 km. The simulated period basically followed the observation time, which covered October 2014 – February 2015 and October 2017 – February 2018. A one-month spin-up was applied in each simulation.

The WRF model is driven by the National Centers for Environmental Prediction Final Analysis (NCEP-FNL) reanalysis data, which then provided the meteorological fields for the CMAQ model. We used CB05 and AERO6 as the gas and particulate matter chemical mechanisms, respectively. The in-line windblown dust and photolytic rate calculation modules were also adopted to improve the simulation. The chemical initial and boundary conditions originated from the interpolated outputs of the Goddard Earth Observing System with chemistry (GEOS-Chem) model (Bay et al., 2001).

The anthropogenic emission inventory for Beijing was taken from the Beijing Municipal Environmental Monitoring Center (BMEMC), which was documented and analyzed in Cheng et al. (2019), while the emission inventory outside Beijing was provided by the Multi-resolution Emission Inventory for China (MEIC) (http://www.meicmodel.org; Zheng et al., 2018) and the MIX emission inventory for the other Asian countries (M. Li et al., 2017). The biogenic emissions were obtained by the Model of Emission of Gases and Aerosols from Nature (MEGAN v2.1); however, open biomass burning was not considered in this work. Detailed model configurations and validations can be found in Cheng et al. (2019), and the simulated results well reproduced the temporal and spatial distributions and variations in $PM_{2.5}$ in Beijing and its surrounding areas. The average simulated $PM_{2.5}$ in Beijing decreased from 91.5 (winter of 2014) to 52.5 (winter of 2017) μg m$^{-3}$, with a total decrease of 39 μg m$^{-3}$, while the observed $PM_{2.5}$ varied from 81.9 to 40.6 μg m$^{-3}$, decreasing by 41.3 μg m$^{-3}$. The Pearson correlation coefficients (R) between the simulated and observed $PM_{2.5}$ in Beijing were 0.81 (winter of 2014) and 0.78 (winter of 2017).

We designed six simulation cases to investigate the impact of meteorological and emission variations. Two base cases were driven by the actual emission inventory and meteorological conditions in the winter of 2014 (case *A*) and winter of 2017 (case *B*). Cases *C* and *D* were designed to quantify the impact of meteorological changes; case *C* was simulated with the emissions in 2014 and meteorological conditions of 2017, while case *D* used the 2017 emissions and 2014 meteorological conditions. Therefore, the differences between *A* and *C* or between *B* and *D* show the influence of meteorological conditions, and the differences between *A* and *D* or between *B* and *C* correspond to the contributions of emission variations. We used the averaged differences as the final impacts. Cases *E* and *F* were developed to evaluate the effect of regional transport on $PM_{2.5}$ variations in Beijing in the winter of 2014 (*E*) and winter of 2017 (*F*). In these two cases, the emissions in Beijing were set to zero, while the regional emissions remained at the actual level. The balances between *A* and *E* or between *B* and *F* represent the contributions of regional transport to the $PM_{2.5}$ concentration in Beijing during the corresponding periods.

## 2.4 Clustering analysis of back trajectories

The Hybrid Single Particle Lagrangian Integrated Trajectory (HYSPLIT) model was conducted to calculate the back trajectories of air masses arriving in Beijing during the observation periods in 2014 and 2017. The meteorological input was downloaded from the National Oceanographic and Atmospheric Administration (NOAA) Air Resource Laboratory Archived Global Data Assimilation System (GDAS) (ftp://arlftp.arlhq.noaa.gov/pub/archives/). Each trajectory was run for three days, with a time resolution of 1 hour, and the initialized height was 100 m above ground level. In total, 2108 and 1292 trajectories were obtained for the winters of 2014 and 2017, respectively. Based on the built-in clustering calculation, the trajectories were then classified





into different groups to represent the main airflows influencing the receptor site. Finally, the optimal 5-cluster and 7-cluster
solutions were adopted for the winters of 2014 and 2017, respectively. Details are shown in Fig. S3.

**2.5 ISORROPIA-II equilibrium calculation**

The ISORROPIA-II thermodynamic model was used to investigate the effects of particle chemical composition on aerosol
properties, i.e., particle pH, AWC, and the partitioning of semivolatile species (Fountoukis and Nenes, 2007). The model computes
the equilibrium state of an $NH_4^+$-$SO_4^{2-}$-$NO_3^-$-$Cl^-$-$Na^+$-$Ca^{2+}$-$K^+$-$Mg^{2+}$-$H_2O$ inorganic aerosol system with its corresponding gases
(Fountoukis and Nenes, 2007). When running the ISORROPIA-II model, it is assumed that the bulk $PM_1$ or $PM_{2.5}$ properties have
no compositional dependence on particle size, and aerosols are internally mixed and composed of a single aqueous phase. The
validity of these assumptions has been evaluated by a number of studies in various locations (Guo et al., 2015; Weber et al., 2016;
M.X. Liu et al., 2017; Li et al., 2018).
The model was run in the forward mode by assuming that aerosol solutions were metastable. Particle water associated with OA
was not considered in this study given its minor effects. M. X. Liu et al. (2017) showed that organic matter (OM)-induced particle
water accounted for only 5% of the total AWC in Beijing. In this study, the transition in aerosol composition was mainly reflected
in the variations in nitrate and sulfate concentrations. For the analysis of the sensitivity of aerosol properties to particle composition,
a selected sulfate concentration combined with the average temperature, RH, and total ammonia concentration ($NH_3 + NH_4^+$)
during the winters of 2014 and 2017 was input into the ISORROPIA-II model, where the total nitrate concentration ($HNO_3 + NO_3^-$)
was left as the free variable.

**3 Results and discussions**

**3.1 Overall variations in aerosol characteristics from 2014 to 2017**

Figures S4 and S5 display the temporal variations in meteorological parameters, trace gases, and aerosol species during the two
winter campaigns, with the average values shown in Table 1. Compared to the frequently occurring haze episodes in the winter of
2014, more clean days with lower $PM_1$ concentrations were observed in the winter of 2017. On average, the $PM_1$ concentrations
were 66.2 μg m$^{-3}$ and 33.4 μg m$^{-3}$ during the winters of 2014 and 2017, respectively. The large reduction in $PM_1$ concentration
reflects the effectiveness of pollution abatement strategies. Satellite-derived estimates also showed an evident decrease in $PM_{2.5}$
concentration in North China in recent years (Gui et al., 2019).

**3.1.1 Changes in SIA characteristics**

Sulfate, nitrate, and ammonium are the dominant components in SIAs and are generally recognized as ammonium sulfate and
ammonium nitrate in $PM_{2.5}$. With the implementation of clean air actions, sulfate underwent the largest decline in the mass
concentration among all SIA species (from 7.7 μg m$^{-3}$ to 2.8 μg m$^{-3}$ during 2014-2017). The contributions of nitrate and ammonium
to $PM_1$ mass reduction were 1.3 μg m$^{-3}$ and 1.5 μg m$^{-3}$, respectively. Different changes in the mass concentration of SIA species
led to variations in the $PM_1$ chemical composition. As illustrated in Fig. 1, nitrate exhibited an increasing mass fraction in $PM_1$
from 18% to 30%, whereas the mass contribution of sulfate decreased from 12% to 8%. Correspondingly, the mass ratio of
nitrate/sulfate increased from 1.4 in 2014 to 3.5 in 2017. Based on the measurements in Beijing from November to December, Xu
et al. (2018) also observed a higher nitrate/sulfate ratio in 2016 (1.36) than in 2014 (0.72). Similar annual variations in aerosol
chemical composition were found in North America over 2000-2016, with an increased proportion of nitrate and a decreased
contribution of sulfate (van Donkelaar et al., 2019). The diurnal cycles of SIAs are displayed in Fig. 2. All SIA species showed



similar diel trends in the two winters, with increasing concentrations after noon due to enhanced photochemical processes and peak
concentrations at night caused by a lower boundary layer height. However, the absolute variations in the SIA mass concentration
differed greatly between 2014 and 2017. While the mass concentration of sulfate decreased by a factor of 2-3 in 2017, nitrate and
ammonium showed much smaller reductions of 15-40% in their mass concentrations throughout the day.
Previous studies have concluded that the dramatically enhanced contribution of sulfate was a main driving factor of winter haze
pollution in China (Wang et al., 2014; Wang et al., 2016; H. Y. Li et al., 2017). However, with the emission mitigation efforts, the
role of SIA species in aerosol pollution changed significantly. Aerosol pollution was classified into three categories in this study:
clean ($PM_1 \leq 35$ µg m$^{-3}$), slightly polluted ($35 < PM_1 \leq 115$ µg m$^{-3}$), and polluted ($PM_1 > 115$ µg m$^{-3}$). The contributions of different
pollution levels and the $PM_1$ chemical compositions at each pollution level are shown in Fig. 3 for the winters of 2014 and 2017.
While the polluted level accounted for 38% of the observation period in the winter of 2014, only 14% of the observation period
was recognized as being polluted in the winter of 2017. In 2014, the mass fraction of sulfate in $PM_1$ was 16.1% during clean
periods. With the increase in pollution level, the contribution of sulfate increased from 10.6% in slightly polluted periods to 13.6%
in polluted periods, while the mass fraction of nitrate decreased. In contrast, sulfate comprised a minor fraction of haze development
in 2017. It was nitrate that exhibited a substantially increased mass fraction at higher $PM_1$ loadings. From clean to polluted periods,
the nitrate contribution to $PM_1$ increased from 22.6% to 34.9%. These results demonstrated that aerosol pollution in Beijing has
gradually changed from sulfate-driven to nitrate-driven in recent years.

### 207  3.1.2 Changes in OA characteristics

In response to the strict emission controls, the mass concentration of organics declined by ~18.5 µg m$^{-3}$ from 2014 to 2017, which
was mainly caused by OOA (~6.8 µg m$^{-3}$) and CCOA (~6.0 µg m$^{-3}$). The contribution from HOA was 2.6 µg m$^{-3}$, which was
associated with the strengthened controls on vehicle emissions. BBOA decreased by 3.2 µg m$^{-3}$ because the use of traditional
biofuels, such as wood and crop residuals, was forbidden in Beijing by the end of 2016. Generally, the concentrations of all OA
factors declined substantially throughout the day in 2017. For primary factors, the reductions in their mass concentrations were
much higher at night than during the day (Fig. 2). Compared to 2014, CCOA decreased by a factor of 4-5 at night in 2017 and a
factor of 1.5 during the day.
Overall, the mass fraction of organics in $PM_1$ declined from 49% to 36% over the period (Fig. 1). The source apportionment results
demonstrated that coal combustion was largely accountable for the reduced contribution of organics. During 2014-2017, the mass
fraction of CCOA in the total OA decreased from 27% to 18%. Reports from the Beijing Municipal Environmental Protection
Bureau (MEPB) also revealed that the contribution of coal combustion to aerosol pollution showed a large decrease during 2013-
2017. The decline in CCOA was largely driven by the reduced emissions of organics from coal combustion with the implementation
of clean air actions. In contrast, the mass contribution of OOA in the total OA increased from 41% to 49% during 2014-2017.
OOA is formed in the atmosphere through various oxidation reactions of volatile organic compounds (VOCs). From 2013 to 2017,
VOCs emissions decreased by approximately half in Beijing but remained constant in the surrounding regions. Large amounts of
OOA brought to Beijing via regional transport weakened the efforts of local emission cuts. Therefore, stronger emission controls
of VOCs need to be placed in both local Beijing and adjacent areas in the future.

### 225  3.2 Factors affecting aerosol characteristics from 2014 to 2017

### 226  3.2.1 Meteorological conditions

To evaluate the influence of weather conditions on air quality improvement, we compared the daily changes in meteorological
parameters during the winters of 2014 and 2017 (Fig. S6). Compared to 2014, the temperature in 2017 was slightly lower





throughout the whole day, which may have facilitated gas-particle conversion for semivolatile species, such as ammonium nitrate.
Although the RH was similar between 2014 and 2017 during the daytime, the nighttime RH in 2017 was slightly higher than that
in 2014, which was favorable for the heterogeneous reactions of secondary species. On average, the observed RH was 29.6% in
the winter of 2014 and 33.9% in the winter of 2017. Diurnal cycles of WS showed that the WS in winter of 2017 was somewhat
higher, implying beneficial conditions for the dispersal of air pollutants. To illustrate the variations in WD, the observed data were
classified into four groups: from north to east (N-E; $0° \leq WD < 90°$), east to south (E-S; $90° \leq WD < 180°$), south to west (S-W;
$180° \leq WD < 270°$), and west to north (W-N; $270° \leq WD \leq 360°$). As displayed in Fig. S6d, the winters of 2014 and 2017 were
both dominated by W-N and N-E, which usually bring clean air masses. After noon, the contribution of winds from S-W started
to increase. According to previous studies, southerly winds arriving in Beijing generally carry higher levels of air pollutants from
the southern regions (Sun et al., 2006; Zhao et al., 2009).
Simulations with the WRF-CMAQ model helped to assess the relative importance of meteorology for changes in aerosol
concentration and chemical composition. The effects of meteorology were quantified by comparing cases *A* and *C* or cases *B* and
*D*. The differences between *A* and *D* or *B* and *C* reflected the effectiveness of emission control. For the total $PM_{2.5}$ concentration,
the simulation results clearly demonstrated that variations in meteorology from 2014 to 2017 had a much lower influence on the
$PM_{2.5}$ reduction than the changes in air pollutant emissions (Fig. S7). On average, changes in weather conditions resulted in a $PM_{2.5}$
decrease of 9.6 $\mu g\ m^{-3}$, which explained 24.8% of the total $PM_{2.5}$ reduction. These results suggest that meteorological variations
are far from sufficient to explain $PM_{2.5}$ abatement during 2014-2017. In terms of aerosol composition, we compared the simulated
results of cases *B* and *D* and found that meteorological changes from 2014 to 2017 had a negligible influence on the chemical
composition of $PM_{2.5}$ (Fig. 4). Therefore, we conclude that weather conditions in 2017 marginally favored air quality improvement
in Beijing, and emission reductions in air pollutants played a dominant role in the variations in aerosol concentration and
composition.

### 3.2.2 Emission changes

According to both the observations (Fig. 1) and simulation results (Fig. 5a), sulfate and organics experienced the largest decreases
among different components in Beijing from 2014 to 2017, which is consistent with the considerable emission reductions in $SO_2$
and primary OC in local Beijing and its surrounding regions (Fig. 6; i.e., Tianjin, Hebei, Shandong, Henan, Shanxi, and Inner
Mongolia). Comparatively, the wintertime nitrate concentration showed the lowest reduction during 2014-2017, which was
expected from the smaller emission cut of $NO_x$ in Beijing and its surrounding areas.
Based on the bottom-up emission inventories (Zheng et al., 2018; Cheng et al., 2019), $SO_2$ emissions decreased by 79.9% in Beijing
during 2014-2017, mainly due to the effective control of coal combustion sources and the optimization of the energy structure.
Until 2017, all coal-fired power units were shut down, and small coal-fired boilers with capacities of <7 MW were eliminated in
Beijing, which reduced coal use by more than 17 million tons. In addition, most of the clustered and highly polluted enterprises
and factories were phased out during this period. These control measures remarkably reduced $SO_2$ emissions from power and
industry sectors. Enhanced energy restructuring was also implemented in the residential sector. During 2013-2017, more than 2
million tons of residential coal was replaced by cleaner natural gas and electricity, involving 900,000 households in Beijing. Apart
from coal burning, the use of traditional biomass, such as wood and crops, was thoroughly forbidden in Beijing by the end of 2016.
The strict governance of residential fuel also made substantial contributions to the BC and OC emission reductions in Beijing,
which decreased by 71.2% and 59.9%, respectively, during 2014-2017. In comparison, $NO_x$ showed a lower emission reduction
of 38.1% from 2014 to 2017 in Beijing. The decline in $NO_x$ emissions was mainly caused by the strengthened emission control of
on-road and off-road transportation, the shutdown of all coal-fired power plants, and the application of low-nitrogen-burning (LNB)



technologies in industrial boilers. However, due to the insufficient end-of-pipe control of widespread gas-fired facilities and the
rapid increase in the vehicle population (the number of vehicles in Beijing increased by nearly 10% during 2013-2017), the $NO_x$
emission reduction in Beijing was not as significant as the $SO_2$ emission reduction.
In adjacent regions, $SO_2$ emissions decreased by 50.6% from 2014 to 2017, while $NO_x$ emissions showed a much smaller reduction
of 15.2%. Comparatively, the energy structure adjustments in surrounding areas were less intense than those in Beijing. Emission
reductions in $SO_2$ and $NO_x$ in surrounding regions were mainly attributed to ultralow power plant emissions and the reinforced
end-of-pipe control of key industries. Because of the looser emission standards for vehicles and the lack of vehicle management,
control measures on transportation in adjacent regions were highly insufficient for $NO_x$ emission reduction compared with those
in Beijing. Overall, the observed transition in $PM_1$ chemical composition with increasing nitrate contribution and decreasing sulfate
fraction was in agreement with the emission changes in their precursors.
**3.2.3 Regional transport**
Variations in regional weather patterns and emission changes in air pollutants in surrounding regions influenced the effect of
regional transport on aerosol characteristics in Beijing. Statistical analysis of air mass trajectories was performed using the
HYSPLIT model. Based on the clustering technique, back trajectories were classified into groups of similar length and curvature
to identify the main airflows affecting the site. The five-cluster solution and seven-cluster solution were adopted for the winters of
2014 and 2017, respectively. The $PM_1$ mass concentration and mass composition for each cluster are shown in Fig. S8. For a better
comparison between 2014 and 2017, clusters were further grouped into two categories according to $PM_1$ loadings. Clusters arriving
in Beijing when the local $PM_1$ concentration was less than 35 $\mu g\ m^{-3}$ were recognized as clean clusters, while clusters with $PM_1$
concentrations greater than 35 $\mu g\ m^{-3}$ were defined as polluted clusters. As displayed in Fig. 7, the average $PM_1$ concentration in
local Beijing was 114 $\mu g\ m^{-3}$ in 2014 when the polluted clusters arrived, which was much higher than that in 2017 (74 $\mu g\ m^{-3}$).
While the contribution of polluted clusters in 2014 was 47%, polluted air masses transported from surrounding regions influenced
Beijing approximately 20% of the time in 2017. The results here indicate that compared to 2014, Beijing was less influenced by
polluted air masses transported from surrounding areas in 2017 during the wintertime, which benefited air quality improvement.
In addition, air masses in 2017 brought more nitrate and less sulfate to Beijing than those in 2014.
The WRF-CMAQ model simulations showed that the contributions of regional transport to the $PM_{2.5}$ concentration in Beijing were
31.4 $\mu g\ m^{-3}$ and 19.0 $\mu g\ m^{-3}$ in the winters of 2014 and 2017, respectively (Fig. 5b). Although the proportion of regional transport
(relative to the total $PM_{2.5}$ concentration in Beijing) remained at approximately 35% in the two winters (34.4% in the winter of
2014 and 36.4% in the winter of 2017), the absolute amount decreased by 39.6%. This result further supported that less $PM_{2.5}$
transported from surrounding regions indeed helped with $PM_{2.5}$ abatement in Beijing. Compared with 2014, the variations in $PM_{2.5}$
components due to regional transport (Fig. 5b) in 2017 were basically consistent with the total aerosol composition changes that
were observed (Fig. 1) and simulated (Fig. 5a) in Beijing. Sulfate had the most notable decrease, with a decreasing rate of 57.9%,
and the regional transport of OM and BC decreased by over 38%. The significant reduction in sulfate was mainly attributed to the
effective $SO_2$ emission controls in the surrounding regions, such as the special emission limits for power plants and the innovation
of industrial boilers. The decreasing rate of regional transport OM was obviously lower than the total change, suggesting that the
local emission controls of VOCs and primary OM in Beijing had a dominant contribution to the decrease in OM. The reduction in
nitrate from regional transport was much smaller than that in other components. This was not only due to the insufficient $NO_x$
emission controls in the surrounding areas but also the relatively rich ammonium environment in North China, which might have
weakened the effects of $NO_x$ reductions. Therefore, the collaborative reductions in $NO_x$ and $NH_3$ are important for future air
pollution control strategies (Liu et al., 2019).




### 3.2.4 Formation mechanisms

From a traditional viewpoint, sulfate formation mainly includes $SO_2$ oxidation by OH in the gas phase and $SO_2$ oxidation in cloud droplets by $H_2O_2$ and $O_3$ in the aqueous phase (Seinfeld and Pandis, 2012). This is actually the case for global sulfate production (Roelofs et al., 1998). The formation rate of sulfate through aqueous reactions is typically much faster than that through gas-phase oxidations. Recently, studies have found that the heterogeneous oxidation of $SO_2$ in aerosol water, which is usually ignored in current model simulations, plays an important role in the persistent formation of sulfate during haze events in China (B. Zheng et al., 2015; Cheng et al., 2016; Wang et al., 2016). However, with the substantial decrease in $SO_2$ emissions currently, the importance of heterogeneous chemistry in sulfate formation is highly uncertain.

To shed light on this query, the formation of sulfate and nitrate with increasing RH was compared between 2014 and 2017. As displayed in Fig. 8, the $SO_4$/BC ratio was much lower in 2017 than in 2014, especially at a higher RH, indicating greatly weakened sulfate formation in 2017 compared to primary BC emissions. $NO_3$/BC showed little difference between 2014 and 2017. The oxidation ratios of sulfur and nitrogen were further estimated as SOR (the molar ratio of sulfate to the sum of sulfate and $SO_2$) and NOR (the molar ratio of nitrate to the sum of nitrate and $NO_x$), respectively. Median values were used for comparison between 2014 and 2017 to avoid bias caused by outliers. When the RH>50%, SOR started to increase significantly with the enhancement in RH in 2014, which was consistent with previous observations in Beijing in 2013 (G. J. Zheng et al., 2015). A year-long study in Beijing from 2012 to 2013 also revealed that a rapid increase in SOR was found at a RH threshold of ~45% (Fang et al., 2019). However, the starting point of SOR growth was clearly delayed in 2017, with a higher RH of 70%. Considering the decrease in the $SO_2$ mixing ratio from 15.5 ppb in the winter of 2014 to 2.8 ppb in the winter of 2017 (Table 1), the results here imply that with the large reduction in gaseous precursors, the rapid formation of sulfate through heterogeneous reaction is more difficult to occur. In addition to emission reduction, reduced regional transport from southern polluted regions in 2017 helped to lower $SO_2$ concentrations in Beijing. Previous studies have revealed the positive feedback between aerosols and boundary layers, as high aerosol loadings could decrease the boundary layer height and further increase aerosol concentrations (Petäjä et al., 2016; Z. Li et al., 2017). With a lower $PM_{2.5}$ concentration in 2017, the interactions between aerosols and the boundary layer were weakened, which in turn also favored a decrease in the $SO_2$ concentration. At a lower RH, the SOR in 2017 (~0.14) was unexpectedly higher than that in 2014 (~0.06), demonstrating a higher sulfate production rate in 2017. Similar results have been observed over the eastern United States, where a considerable decrease in $SO_2$ resulted in the more efficient formation of particulate sulfate during wintertime (Shah et al., 2018). Combining airborne measurements, ground-based observations, and GEOS-Chem simulations, Shah et al. (2018) explained that sulfate production in winter is limited by the availability of oxidants and particle acidity. At lower concentrations of precursor gases, the oxidant limitation on $SO_2$ oxidation weakened, leading to a higher formation rate of sulfate. Particulate nitrate in $PM_{2.5}$ is mainly formed through the neutralization of $HNO_3$ with $NH_3$. $HNO_3$ is produced by $NO_2$ oxidation via OH during the day and the hydrolysis reaction of dinitrogen pentoxide ($N_2O_5$) at night, with the former being the dominant pathway (Alexander et al., 2009). At a lower RH, NOR was slightly higher in 2017 than in 2014 (Fig. 8), which may have been caused by the reduced limitation of oxidants with lower $NO_x$ emissions in 2017. Nitrate formation was also affected by the competition for available $NH_3$ between sulfate and nitrate. In the atmosphere, $NH_3$ prefers to react first with $H_2SO_4$ to form ammonium sulfate due to its stability. When excess $NH_3$ is available, ammonium nitrate is formed (Seinfeld and Pandis, 2012). With the decrease in sulfate concentration in 2017, some $NH_3$ was freed up to react with $HNO_3$. This may have also facilitated the formation rate of nitrate. When RH>60%, NOR increased substantially in 2014 and 2017, indicating the importance of heterogeneous reactions in nitrate production.





**3.3 Influence of the transition in aerosol characteristics on particle properties**

According to thermodynamic calculations, various aerosol properties were affected by changes in aerosol characteristics associated with clean air actions. As shown in Fig. 9a, nitrate and sulfate play key roles in determining the AWC in $PM_{2.5}$. The decreasing mass concentrations of nitrate and sulfate result in a lower AWC. Similar observations have been reported previously across northern China, revealing that nitrate and sulfate are dominant anthropogenic inorganic salts driving AWC (Wu et al., 2018). With the clean air actions enacted, the mass concentrations of nitrate and sulfate decreased during 2013-2017, leading to an average decline in AWC from 4.8 to 2.5 µg m$^{-3}$. Data for the winter of 2013 were acquired from Sun et al. (2016). The reduced AWC further helped air quality improvement by lowering the ambient aerosol mass and enhancing visibility. Because aqueous-phase reactions contribute largely to sulfate formation in winter, the decrease in AWC decelerated the formation of sulfate. In addition, the lower AWC slowed down the uptake coefficient of $N_2O_5$ for heterogeneous processing, thereby suppressing the formation of particulate nitrate.

Figure 9b displays the effects of nitrate and sulfate concentrations on particle acidity. Particle acidity is largely driven by the mass concentration of sulfate and is less sensitive to the variation in nitrate. Particle pH substantially decreases with increasing sulfate concentration. In contrast, more particulate nitrate leads to a slightly higher pH by increasing the particle liquid water and diluting aqueous H$^+$ concentrations. Through the comparison of pH predictions among various locations worldwide, Guo et al. (2018) also found that a higher particle pH was generally associated with higher concentrations of nitrate. During 2013-2017, the average particle pH varied from 5.0 to 6.2, with a significant decrease in sulfate concentration, resulting in a more neutral atmospheric environment. When pH > 5.0, aqueous-phase productions of sulfate are dominated by $SO_2$ oxidation with $H_2O_2$, $O_3$, and $NO_2$ under haze conditions in Beijing (Cheng et al., 2016). The sulfate oxidation rates by $O_3$ and $NO_2$ increase with increasing particle pH. Therefore, a more neutral atmosphere would favor aqueous-phase sulfate formation in Beijing. Particle acidity also influences the gas-particle partitioning of nitrate. The rising particle pH would result in a higher fraction of particulate nitrate ($\in (NO_3^-) = \frac{[NO_3^-]}{[HNO_3]+[NO_3^-]}$). Figure S9a displays the variation in $\in (NO_3^-)$ as a function of particle pH under typical Beijing winter conditions (temperature of approximately 0℃). With a particle pH below 3, $\in (NO_3^-)$ increases sufficiently with the enhancement in particle pH. However, when the particle pH is larger than 3, $\in (NO_3^-)$ remains relatively stable (approaching 1). From 2013 to 2017, with the particle pH remaining above 3 in Beijing, no clear change in $\in (NO_3^-)$ was observed (Fig. S9b).

The variations in nitrate and sulfate concentrations also affected the gas-particle partitioning of total ammonium ($NH_x = NH_3 + NH_4^+$). As expected, the decreased concentrations of nitrate and sulfate led to a reduction in the ammonium particle fraction ($\in (NH_4^+) = NH_4^+/NH_x$; Fig. 10). From 2013 to 2017, $\in (NH_4^+)$ in Beijing always stayed below 0.3, indicating that most ammonium existed in the gas phase. Therefore, a minor reduction in $NH_x$ would not be sufficient for air quality improvement. Guo et al. (2018) revealed that for winter haze conditions in Beijing, an approximate 60% decrease in $NH_x$ was required to achieve an effective reduction in $PM_{2.5}$. Due to the close linkage between ammonia emissions and agricultural activities, it may be difficult to attain substantial ammonia reduction in China.

**4 Conclusions**

This study investigated the variations in aerosol characteristics in Beijing during the winters of 2014 and 2017 by combining the online measurements of aerosol chemical composition with a comprehensive model analysis of meteorological conditions, anthropogenic emissions, and regional transport. The average $PM_1$ concentration decreased from 66.2 µg m$^{-3}$ in the winter of 2014 to 33.4 µg m$^{-3}$ in the winter of 2017, with decreasing concentrations of organics, sulfate, nitrate, and ammonium by 18.5 µg m$^{-3}$, 4.9 µg m$^{-3}$, 1.3 µg m$^{-3}$, and 1.5 µg m$^{-3}$, respectively. These changes reduced the mass fractions of organics and sulfate from 59%





to 36% and from 13% to 9%, respectively, whereas increased the nitrate contribution from 19% to 32%. Consequently, the winter
haze pollution changed from sulfate-driven to nitrate-driven in Beijing from 2014 to 2017, implicating the increasing role of nitrate
in aerosol pollution.
The chemical transport model simulations suggest that the rapidly declining emissions in Beijing and its adjacent regions account
for ~75% of $PM_{2.5}$ abatement in Beijing, and the remaining portion can be explained by the favorable weather conditions in 2017.
The faster reductions in $SO_2$ emissions compared to $NO_x$ emissions are in line with the decreased sulfate contribution and increased
nitrate fraction in observed aerosols, and the model simulations with these emission estimates can reproduce the relative changes
in aerosol composition. Regional transport contributed moderately to the variations in aerosol concentration and its chemical
composition, with less polluted air masses transported from surrounding regions to Beijing in the winter of 2017. The air masses
were observed to have brought more nitrate and less sulfate to Beijing. Furthermore, the considerable decrease in $SO_2$ emissions
suppressed the rapid formation of sulfate during wintertime. The fast $SO_2$-to-sulfate conversion through heterogeneous reactions
was observed to increase promptly at a RH threshold of ~50% in 2014, while a higher RH of 70% was observed in 2017.
Thermodynamic calculations showed that the decreased sulfate and nitrate concentrations in 2017 caused a lower AWC in $PM_{2.5}$,
which further decreased the ambient aerosol mass and weakened the formation rates of sulfate and nitrate through aqueous-phase
reactions. Particle acidity displayed a decline during 2014-2017, mostly driven by the declining sulfate concentration. In turn, the
more neutral ambient environment would favor the aqueous oxidation of sulfate in Beijing. Analysis of the ammonium particle
fraction indicated that most ammonium in Beijing existed in the gas phase. Therefore, increased efforts are needed to achieve an
effective reduction in particle ammonium in the future.

**Author contributions**

QZ and KH conceived the study. HL conducted the field measurements and carried out the data analysis. JC provided the emission
data and performed the model simulations. BZ participated the data analysis. HL, JC and QZ wrote the paper with inputs from all
coauthors.

**Acknowledgements**

This work was funded by the National Natural Science Foundation of China (41571130035, 41571130032 and 41625020).

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





**Table 1** Summary of the average meteorological parameters, mixing ratios of gaseous species, and mass concentrations of the PM$_1$ chemical
components during the winters of 2014 and 2017.

| Sampling period | | 2014 winter | 2017 winter |
|---|---|---|---|
| Meteorological parameters | T (°C) | 1.70 | -2.26 |
| | RH (%) | 29.6 | 33.9 |
| | WS (m s$^{-1}$) | 1.58 | 1.73 |
| Gaseous species | SO$_2$ (ppb) | 15.5 | 2.8 |
| | NO$_2$ (ppb) | 26.0 | 24.9 |
| | CO (ppm) | 1.6 | 0.7 |
| | O$_3$ (ppb) | 14.4 | 15.5 |
| Aerosol species (μg m$^{-3}$) | Org | 30.4 | 11.9 |
| | HOA | 4.1 | 1.5 |
| | BBOA | 5.6 | 2.4 |
| | CCOA | 8.2 | 2.2 |
| | OOA | 12.6 | 5.8 |
| | SO$_4$ | 7.8 | 2.8 |
| | NO$_3$ | 11.2 | 9.9 |
| | NH$_4$ | 6.9 | 5.4 |
| | Chl | 3.4 | 1.7 |
| | BC | 2.4 | 1.5 |
| | PM$_1$ | 66.2 | 33.4 |






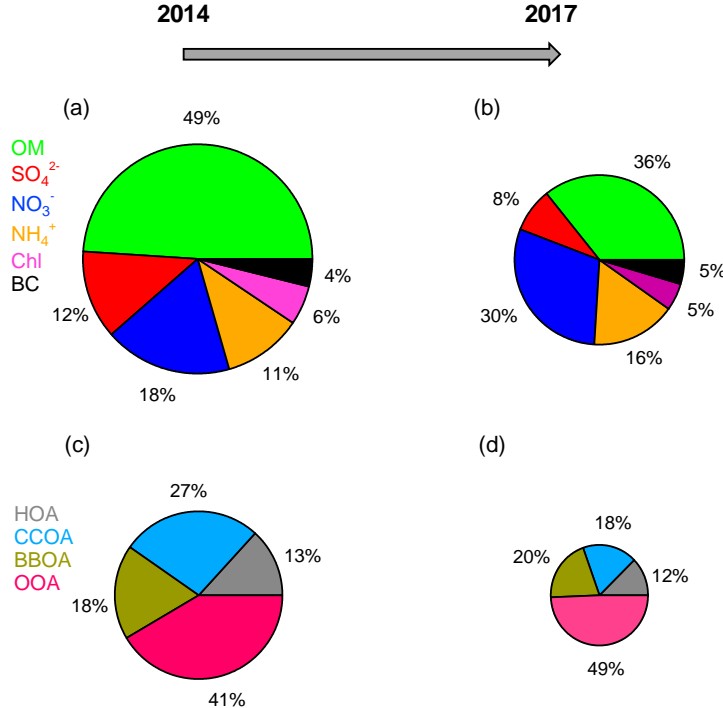


**Figure 1. Average chemical compositions of PM₁ and OA in (a, c) winter of 2014 and (b, d) winter of 2017. The decreasing rates of different components from 2014 to 2017 are as follows: 60.9% for organics, 64.1% for sulfate, 11.6% for nitrate, 21.7% for ammonium, 50.0% for chloride, and 37.5% for BC.**





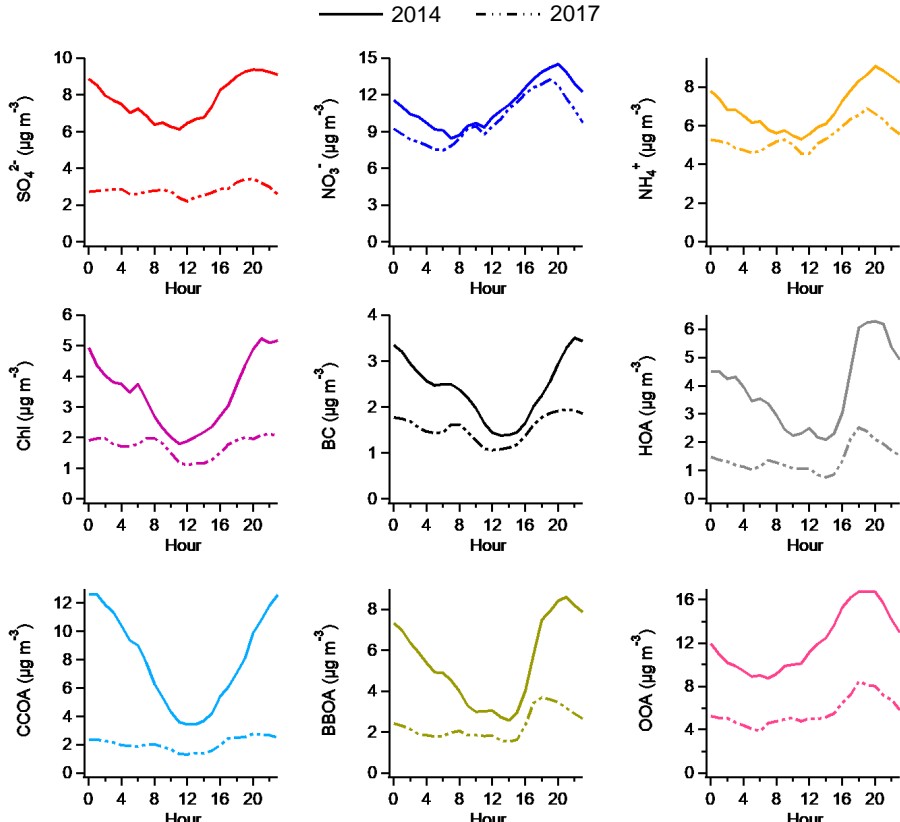


**Figure 2. Average diurnal cycles of different aerosol species in the winter of 2014 (solid line) and winter of 2017 (dashed line).**



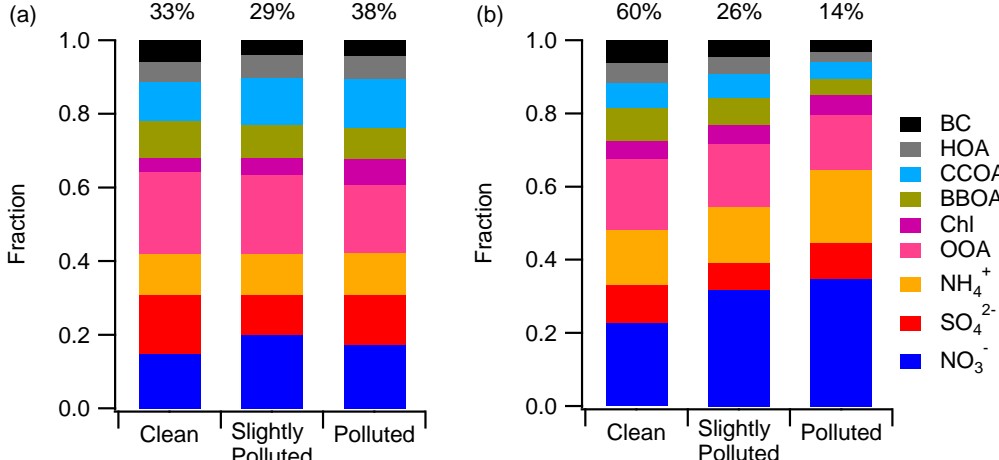

**Figure 3. Aerosol chemical composition at different pollution levels in the (a) winter of 2014 and (b) winter of 2017. The contributions of each pollution level are shown at the top of each bar.**





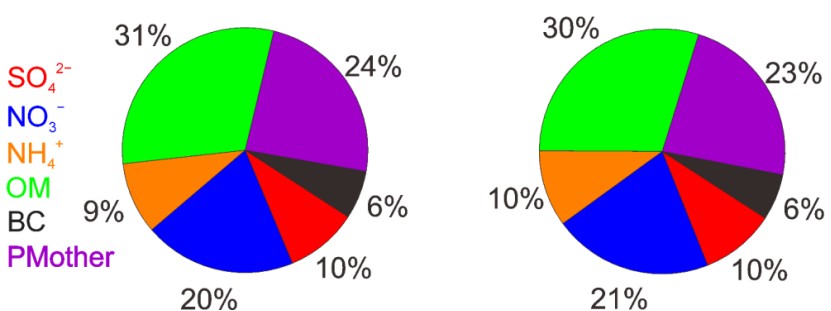


**Figure 4. The average PM$_{2.5}$ chemical composition simulated by the WRF-CMAQ model for the observation periods in 2017: (a) base scenario with the 2017 emissions and the 2017 meteorological conditions; (b) simulation with the 2017 emissions and 2014 meteorological conditions.**




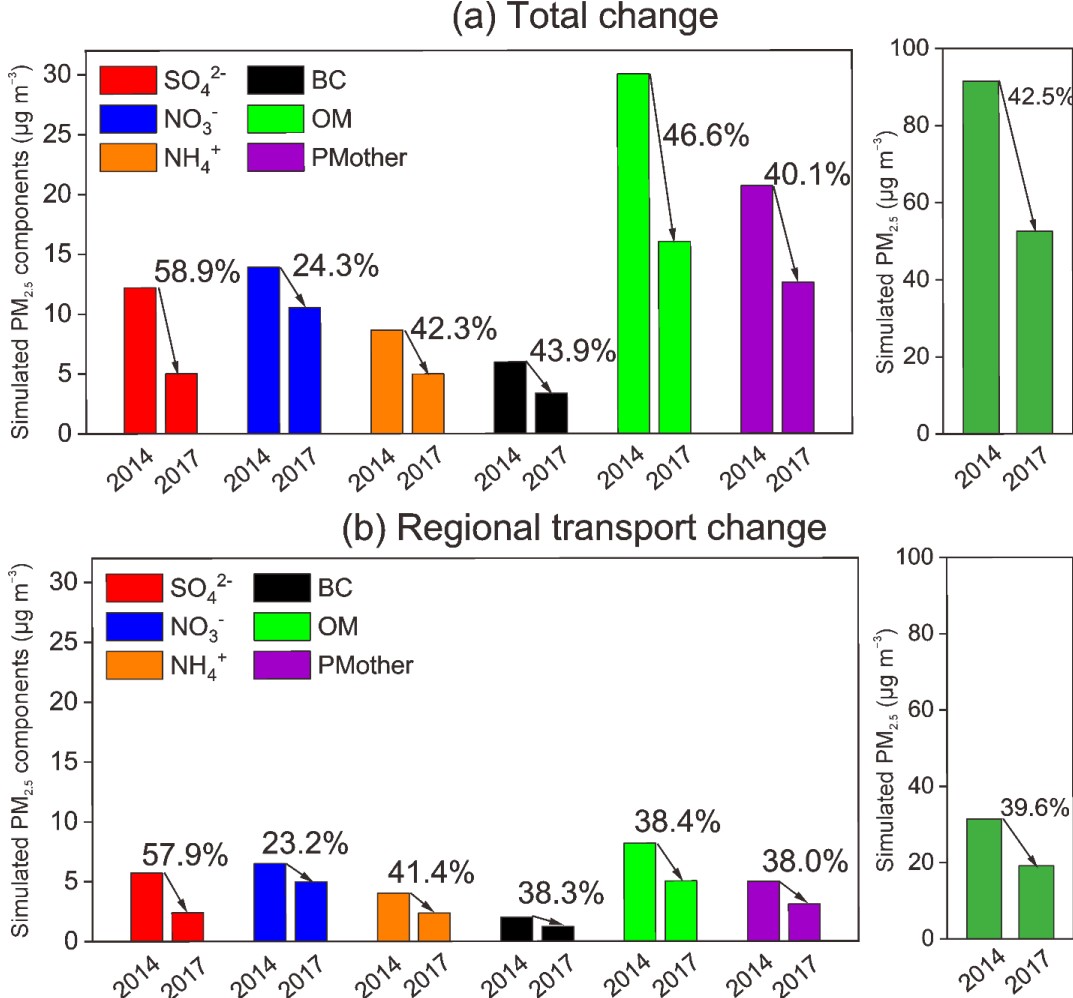


**Figure 5. Simulated concentrations of PM₂.₅ and its chemical components during the observation periods of 2014 and 2017: (a) total**
**changes in Beijing and (b) changes due to regional transport.**





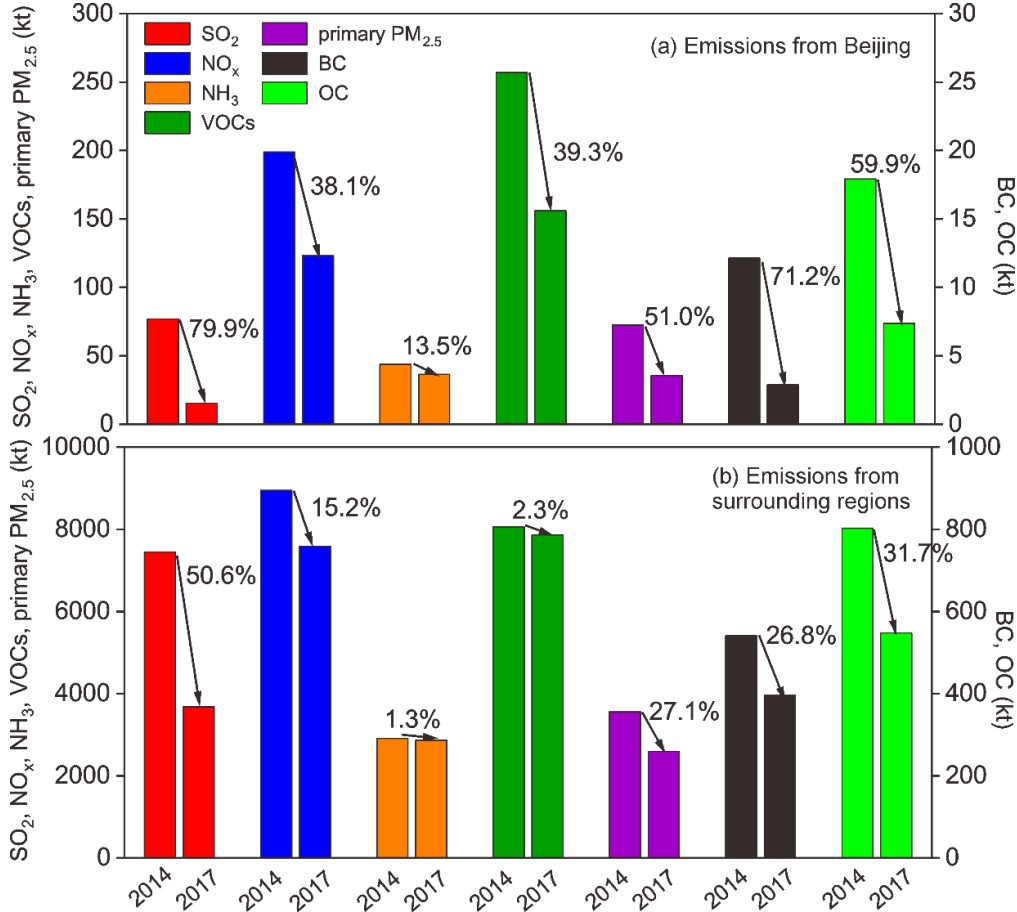


**Figure 6. Changes in the anthropogenic emissions of SO₂, NOₓ, NH₃, VOCs, primary PM₂.₅, BC, and OC in (a) Beijing and (b) its surrounding regions from 2014 to 2017.**




**Figure 7. Comparison of the air masses arriving in Beijing between 2014 and 2017. (a) and (b) show the clustering analysis of the back**
**trajectories in the winters of 2014 and 2017, respectively, with pie charts displaying the contributions of the clean and polluted air masses.**
**The stacked bar charts on the right show the average aerosol compositions for the clean and polluted clusters.**





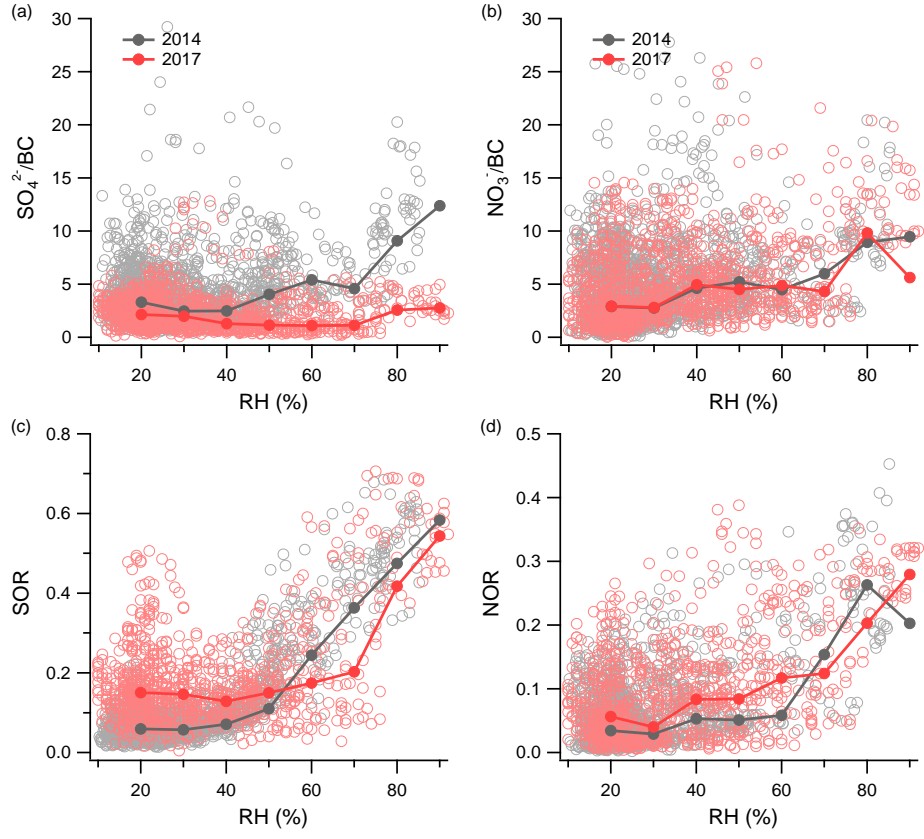


**Figure 8. Variations in (a) SO₄/BC, (b) NO₃/BC, (c) SOR, and (d) NOR plotted against increasing RH. The data are also binned according to RH values, with the median value shown for each bin.**





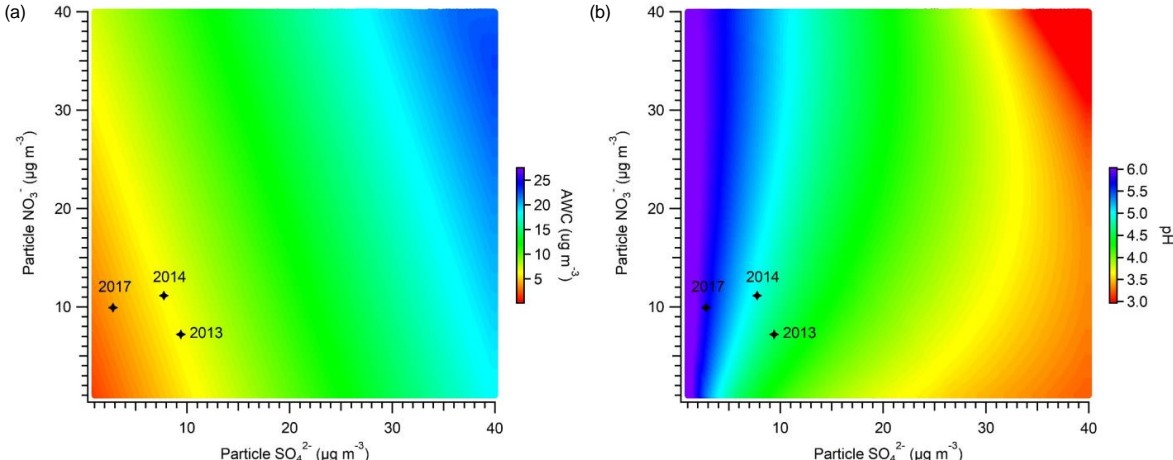


**Figure 9. Sensitivity of (a) AWC and (b) particle pH to the mass concentrations of particulate sulfate and nitrate. The stars indicate the**
**average winter conditions for the years 2013, 2014, and 2017.**



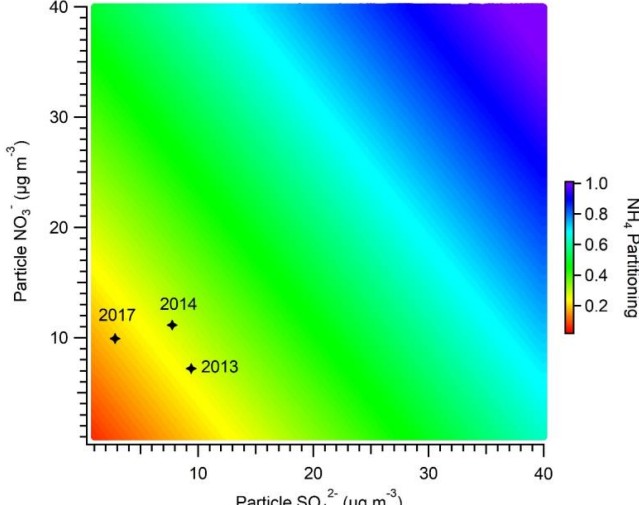


**Figure 10. Sensitivity of the ammonium partitioning ratio to the mass concentrations of particulate sulfate and nitrate. The stars indicate the average winter conditions for the years 2013, 2014, and 2017.**