# Peer review of "Rapid transition in winter aerosol composition in Beijing from 2014 to 2017: response to clean air actions"

_Atmospheric Chemistry and Physics, 2019_

## Referee Comment (RC1) · Rodney Weber (Referee) · 6 Jun 2019

In past publications, the authors have reported on the effects of emission controls on China's air quality. This paper focuses on how those emission changes affect aerosol chemical composition. The authors present a comprehensive analysis involving measurements and models to assess the role of just emissions, apart from transport and meteorological differences. The results are interesting and the topic appropriate for this journal, however some details were missing and in many cases the analysis or description of processes was not of sufficient technical detail, such as the thermodynamic analysis. Also, strong conclusions are made with limited analysis to support them. For example, a major finding seems to be that these changes in emissions altered the aerosol formation mechanisms. It is stated in the Abstract and Conclusions

that lower SO2 suppressed rapid sulfate formation through heterogeneous reactions. But it is not clear that this is really shown or tested in a quantitative way, instead it seems to be mostly speculation, based mainly on this hypotheses being consistent with the observations. Line 324 simply states: the results here imply that. . . Could not a chemical transport model, or maybe a simpler 0-D or 1D box model be run to further test this hypothesis? Overall, my recommendation is that the authors check closely if their reported findings are truly supported in the manuscript by quantitative analyses, and if only based on a consistency with expectations, that this clearly be stated. More detailed comments are provided below.

Specific Comments:

Line 59-60; I would think atmospheric chemical reactions (secondary aerosol) and deposition would also be a major contributor to PM2.5 composition.

Line 136 to 137, in giving the ambient data vs model comparison, state the integration time, ie 24 hr average data?

ISORROPIA calculations: In this paper the model is run without considering non-volatile cations. Maybe a few words should be added why this is ok, ie, it may be argued that for PM1 this is reasonable. As another example, the nitrate considered in the paper is all semivolatile nitrate (ie, NH4NO3), but it is possible that nonvolatile nitrate also exists in the ambient PM2.5 (eg, Ca(NO3)2). Thus if Ca2+ was considered in the thermodynamic calculations, it could affect predicted pH and NO3- concentrations. Most of this Ca2+ would likely be in the 1 to 2.5 um range, and since the comparison between PM1 and PM2.5 mass is reasonable, the authors could argue that it is not a large contribution. Also, I suggest the authors specifically note what RH range was used in the thermodynamic calculations, many of the assumptions, such as no separate organic/inorganic phases, etc, may be less likely at lower RH. (say <40 to 50%). Line 231-232 notes that the observed RH was about 33 to 34% in the winter of 2017. This a very low RH to comfortably run ISORROPIA under the metastable assumption

without some test on the reliability of the results.

In the experimental section, there was no discussion of measurement of HNO3 or NH3 (gas species), yet ISORROPIA was run such that these data are needed, ie, run in forward mode. More information is needed in the paper on how the model was run without these critical gas phase species.

Line 175 or in Table 1 title state this is observational data (not model).

Line 258, what does Until 2017 mean? These changes were completed by 2017?

Line 311, does not tell the complete story. There are publications, see below, that show the predictions of Cheng et al. 2016 and Wang et al, 2016 are likely not correct due to their incorrect calculation of fine particle pH and that this proposed heterogeneous chemistry is highly sensitive to pH. This counter argument should also be noted here in this paper for completeness.

Liu, M., Y. Song, T. Zhoh, Z. Xu, C. Yan, M. Zheng, Z. Wu, M. Hu, Y. Wu, and T. Zhu (2017), Fine Particle pH during Severe Haze Episodes in Northern China, Geophys. Res. Lett., 10.1002/2017GL073210.

Guo, H., R. J. Weber, and A. Nenes (2017), High levels of ammonia do not raise fine particle pH sufficiently to yield nitrogen oxide-dominated sulfate production, Sci. Reports, 7(12109), DOI:10.1038/s41598-41017-11704-41590.

Song, S., M. Gao, W. Xu, J. Shao, G. Shi, S. Wang, Y. Wang, Y. Sun, and M. B. McElroy (2018), Fine particle pH for Beijing winter haze as inferred from different thermodynamic equilibrium models, Atm. Chem. Phys., 18, 7423-7438.

Line 318-319, define SOR and NOR (ie, what does the acronym stand for?)

Lines 339 to 344. Although this discussion tends to follow the Seinfeld and Pandis discussion of sulfate/nitrate/ammonium interactions, it is largely based on weak intuitive arguments and not a rigorous thermodynamic discussion. It is suggested that these

types of statements be avoided. Below are possible other ways to discuss the interplay between sulfate, nitrate and ammonium and gas species, nitric acid and ammonia:

Change: Particulate nitrate in PM2.5 is mainly formed through the neutralization of HNO3 with NH3 to something like: Semivolatile PM2.5 particulate nitrate is formed through the partitioning of HNO3 to the particle phase, which is more favored at higher aerosol pH. pH is affected by gas phase NH3 concentrations, where higher NH3 generally leads to higher pH and so possibly more particulate nitrate.

Change: Nitrate formation was also affected by the competition for available NH3 between sulfate and nitrate. In the atmosphere, NH3 prefers to react first with H2SO4 to form ammonium sulfate due its stability. To: Because sulfate is nonvolatile, when it is a significant fraction of the aerosol mass it has a dominant influence on aerosol pH, making the aerosol acidic (low pH). In contrast ammonium and nitrate are semivolatile and so their particle-phase concentrations depend on the meteorological conditions (T, RH) their corresponding gas phase concentrations, (NH3 and HNO3 respectively) and aerosol pH. For example, at high sulfate and moderate NH3 concentrations the aerosol can be too acidic for partitioning of HNO3, but at higher NH3, or if sulfate concentrations drop sufficiently (or RH increases), particle pH will increase and can reach a point at which HNO3 partitioning can occur and nitrate aerosol formed. Lower T also favors partitioning to the particle phase through Henry's law constants.

As for the last line, when RH>60% maybe an additional explanation for the trend in Fig 8d is that as RH increases liquid water levels increase resulting in higher pH, which allows more nitrate to partition to the aerosol through a feedback loop, as is discussed later in the paper. That is, the increase in NOR may be due to more than just heterogeneous nitrate production.

Line 366-369, In Fig S9a provide a reference for the calculation of epsilon(NO3-). This pH of 3 at which the sensitivity of epsilon(NO3-) changes, as found in this work, was discussed in detail by Guo et al., ACP 2018, which should be cited.

Finally, there are a number of studies reporting pH in Beijing over different periods. Some of these did analysis to test the thermodynamic model predictions, which was not done here. A summary of these papers and comparison to pH reported in this paper is warranted to support this analysis.
* * *

---

## Short Comment (SC1) · 9 Jun 2019

(a) Overview The authors used fine particle data collected in Beijing during the winters of 2014 and 2017 to understand changes in the chemical composition of the PM over a period when particle mass decreased significantly. The results here are consistent with past data, showing a large decrease in PM mass, but also revealing that this is largely because of decreases in sulfate, organics, and unidentified components, as a result of decreasing local and regional emissions. The authors find that meteorology also contributed to lower PM concentrations in 2017 but that this was a smaller factor.

This is an interesting paper with a number of complementary components. On the other hand, there are a number of issues that need to be addressed, as described

below. The thermodynamic modeling seems problematic and needs more clarification and information.

(b) Major Points 1a. Figure 2 and lines 190 – 191: The text states that "All SIA species showed similar diel trends in the two winters...". For some species this is true, but for many of the species, the diel variability is much lower in 2017 than in 2014, suggesting that while more local sources dominated in 2014, regional sources are more important in 2017. Take CCOA as an example: in 2014, the concentration varied by a factor of 3, while in 2017 the variation was probably only a factor of 1.5 or less. Can the change in diel variability be used as an indicator of local vs. regional pollution?

1b. Of course boundary layer height is a major influence on concentrations as well. I suggest you add a plot of CO as a tracer for BL height. Does plotting the ratio of PM component/CO help disentangle chemistry and BL height? These might be useful supplemental plots.

2. Figure 8 and its corresponding text. (i) Why normalize panels (a) and (b) to BC concentrations? Since both the numerator and denominator are changing between 2014 and 2017, the ratio seems less effective as a normalizing factor. What do the plots look like if sulfate and nitrate are not normalized by BC? (ii) The data are very noisy and so the authors have chosen to plot the median values (as the solid circles). But what does it look like if the mean and standard deviation are plotted instead? Are there any significant differences between RH values for a given year? If the authors want to stay with median, they should at least show some interquartile ranges or other measures of variability about the median. (iii) Line 323: "...the starting point of SOR growth was clearly delayed in 2017...". Given the spread in the data, this is speculative since the median has no associated uncertainty or variability. Can this difference be tested statistically? (iv) Lines 330 – 335. The authors explain that the higher sulfate / (sulfate + SO2) ratio (i.e., SOR) in 2017 at low RH results from less of an oxidant limitation. But isn't an alternative explanation that a higher fraction of the sulfur pollution in Beijing is from regional transport, which would have a higher SOR since it is more aged? (v) The nitrate / (nitrate + NO2) ratio (i.e., NOR) discussion in lines 339 – 344 seems to be missing a few points. First, NOR is a poor measure for the extent of NOx oxidation since it considers only particulate nitrate and not gas-phase HNO3. Under acidic particle conditions (as seems typical in 2014), most N(V) will be gas-phase HNO3, a result of sulfate acidity driving out the particle nitrate. But this is not accounted for if NOR is calculated only with particulate nitrate, as is done here. Running the e-AIM thermodynamic model on the average 2014 data shows that the gas-phase HNO3 concentration is over an order of magnitude higher than particulate nitrate. Thus N(V) / NOx + N(V), with both phases considered for N(V), would be a better measure of NOx oxidation than NOR. ISOROP-PIA could be run to examine HNO3(g) in the two years. (vi) As a consideration for future work, it would be enormously helpful to have gas-phase measurements of NH3 and/or HNO3 to constrain the modeled aerosol pH.

3. Figure 9 and its corresponding text. (i) This figure indicates that the aerosol pH in 2014 was approximately 5, but this is inconsistent with the average composition indicated in Table 1. Using the averages for 2014 in Table 1, assuming protons make up the missing positive charge (which I assume was also done in ISORROPIA) and using a relative humidity of 60% in the e-AIM model results in a pH of -0.04 assuming a proton activity coefficient of 1; the pH is -0.7 if the e-AIM activity coefficient (4.22) is used. These results were done allowing solids to form, but it shouldn't change significantly if solids were not allowed to form. While the pH undoubtedly varies between samples, it is hard to believe the typical pH is near 5 given the ion imbalance in the data. (ii) Is the range of pH values given in line 361 (pH 5.0 to 6.2) from their work? Why is it for 2013 – 2017, rather than just the two winters of this study? It's not clear that the authors did much with the ISORROPIA data: why not report ALW and pH for every sample? This would be very useful information. (ii) The 2017 average data in Table 1 has a net positive charge, requiring a nonsensical negative concentration for protons that is of similar magnitude (though smaller) than the proton concentration from the 2014 data. This prevents use of a thermodynamic model and suggests that there are other charged species, likely organics, which are affecting the ion balance and the pH

calculation. This should be discussed. Were the authors able to calculate ALW and pH values for the 2017 data? (iii) Under the acidic conditions I calculated for 2014, the value of NO3- / (HNO3 + NO3-) is very small, in contrast to the values described in the manuscript. (v) Similarly, I calculate a NH4+/NHx ratio near unity in 2014, in contrast to the small values reported in the text.

(c) Minor Points 1. Throughout the manuscript, it would be much better (and more consistent with the other inorganic nomenclature in the paper) to use "Cl–" as the abbreviation for chloride rather than "Chl".

2. lines 183-184: This sentence is unclear. Clearer wording would be "The decreases in particulate nitrate and ammonium during this period were 1.3 and 1.5 $\mu$g m–3, respectively."

3. Figure 1. The second line of the figure caption is unclear. These are not "rates", but rather decreases in concentrations, yes? Also, three significant figures in the decreases seems one too many.

4. line 188: It is odd that the Xu nitrate/sulfate ratios are approximately half of the values in the current work. Why this discrepancy?

5. lines 203-204: Sulfate is a minor fraction of haze in 2014 as well as 2017, so this statement isn't very useful. It would be better to indicate the sulfate contributions to each of the pollution classifications in 2017.

6. lines 204–205: This description of nitrate contribution is for 2017?

7. line 208: State the 2014 and 2017 OA concentrations along with the overall decline.

8. line 209: "The contribution from HOA was 2.6 $\mu$g m–3...". This is not as clear as it should be: better to state something like "HOA decreased by 2.6 $\mu$g m–3...".

9. lines 240 – 241: It is not clear what cases A, B, C, and D refer to. Are these panels of Fig. S7?

10. lines 258-259: This is unclear. Should "Until 2017..." be "By 2017..."?

11. line 298: The 57.9% is not a "rate" and likely has one too many significant figure.

(d) Recommendation I recommend that the paper be accepted after major revisions to address the points above.

———————————————

---

## Referee Comment (RC2) · Anonymous Referee #2 · 15 Jul 2019

This study combines in-situ measurements of PM1 composition in Beijing during the winters of 2014 and 2017 and the simulation results from a regional chemical transport model to investigate the impacts that the clean air actions in China have had on aerosol chemistry. The relative contributions of anthropogenic emissions, meteorological conditions, and regional transport to the changes in aerosol composition in Beijing are also investigated. This is an interesting work and provides a timely and relevant analysis of a current problem – how aerosol pollution in Beijing responded to the implementation of the Air Pollution Prevent and Control Policy. Overall, the manuscript is well written and fits well within ACP's aims and scope. However, the current version may require substantial revision before publication can be considered. A major shortcoming in this manuscript is that the Methods section is very much lacking of important

technical details, for example on aerosol source contribution analysis and CMT model performance, thereby raises doubts about the credibility of the results. Moreover, there are some inconsistencies in the results and discussions which could cause concerns about the quality of the data, representativeness of the findings, or validity of the conclusions. Additionally, the figure legends and captions are often too brief to make the figures understandable.

Detailed comments:

Line 44-46, how much did air pollutants reduce between 2013 – 2017 in the Beijing area?

Section 2.2. is cursory and provides very little information on the organic aerosol source apportionment analysis. Details must be provided on how the PMF/ME-2 analysis of the ACSM data was performed, what data treatments were implemented, and how the solution conditions (eg number of factors, a value, fpeak) were selected and evaluated.

In Section 2.3. more information should be provided on the performance of the CMT model at simulating PM2.5 composition and how the modeling results compare to observations for 2014 and 2017 winters separately.

Section 2.5 lists three assumptions about aerosol properties in the ISORROPIA modeling. Several references are cited and claimed to support these assumptions for this study. But upon reading the references more closely, they don't seem so as the references either talked about aerosols from different locations or under different meteorological conditions, or simply did not provide direct evidence on aerosol physical states. In fact, given the wintry weather (very low RH and T) and intense local emissions in Beijing, it is hard to believe that internal mixing and single aqueous phase were the prevailing aerosol conditions relevant to this work.

Line 168 – 171, there was no mentioning of NH3 and HNO3 measurements in 2.1., but

it is mentioned here that the values of NH3+NH4 and NO3+HNO3 were input into the model. Where did the NH3 and HNO3 data come from?

Line 110, missing numbers after "0." What "a" values were used?

Line 167 -168, what does "the transition in aerosol composition" mean in this sentence?

Line 186 – 188, the Xu et al. study was also conducted in Beijing in the winters of similar years, but the nitrate to sulfate ratios reported there were much lower than in this study. Normal measurement uncertainty could not explain such large discrepancies (more than a factor of two in difference). Was it due to measurement artifacts or does it suggest some issues with the representativeness of the measurement data? What's the implication for the validity of the conclusions presented in this paper?

Line 205 – 206, organics were higher than sulfate and nitrate and in both 2014 and 2017, so calling Beijing aerosol pollution being "sulfate-driven" or "nitrate-driven" does not seem logical.

Line 259, for coal usage reduction in addition to quoting the absolute amount, it would be also interesting in knowing the relative amount of reduction.

Figure 2, according to the diurnal profiles , HOA concentration in 2014 was 2-3 times lower than 2017. If HOA is representative of emissions from transportation, is this level of decrease consistent with the decrease in emissions according to emission inventory? Moreover, the decrease of BC concentration from 2014 to 2017 was between 30-40% but the reduction of total combustion POA (sum of CCOA, BBOA and HOA) was close to 70%. This would suggest some very large, probably unrealistic, changes in the combustion emission factors.

Line 289 – 292, the comparisons of PM1 concentrations between different air trajectory classes do not logically lead to a conclusion about how much Beijing aerosol was influenced by polluted air masses transported from surrounding areas. Beijing has local pollution sources which could cause high PM events as well. In fact, in the paragraph

immediately beneath, the authors reported that CMT simulation indicates that regional transport contributes to only 30 -40% of PM2.5 in Beijing.

Section 3.2.4, 2nd paragraph, what are the rationales for using SO4/BC and NO3/BC ratios in the analysis? Sulfate is a secondary species with formation time scales usually much longer than the emission time scales of BC. So the physical meaning of SO4/BC ratio is unclear.

Section 3.2.4, 2nd paragraph, nitate/(nitrate+NOx) is not a proper index for the oxidation ratio of nitrogen. Discussions related to NOR should be either removed or revised.

For the discussions of the relationship between SOR and RH, it is important to point out that RH was measured locally at the sampling site but sulfate was mostly formed on a regional scale, ie, in air masses upwind of Beijing. Is it valid to assume that in-situ RH measurement data are representative of the RH conditions in the air masses where sulfate was formed?

Line 330 -331, does the higher SOR in 2017 than 2014 necessarily demonstrate "a higher sulfate production rate in 2017"? A relatively larger contribution from background air masses could also lead to higher SOR.

Line 544- 545, this citation is incomplete

Figures in the supplementary materials are fuzzy, need to use better resolution.

Figure S6d, explain how to read the figure and the meanings of N-E, W-N, E-S, S-W?

Figure S7, what do the color bars stand for?

---

## Author Comment (AC1) · 29 Jul 2019

**Response to Referee Comment 1 (RC1) on "Rapid transition in winter aerosol composition in Beijing from 2014 to 2017: response to clean air actions" by H. Li et al.**

In past publications, the authors have reported on the effects of emission controls on China's air quality. This paper focuses on how those emission changes affect aerosol chemical composition. The authors present a comprehensive analysis involving measurements and models to assess the role of just emissions, apart from transport and meteorological differences. The results are interesting and the topic appropriate for this journal, however some details were missing and in many cases the analysis or description of processes was not of sufficient technical detail, such as the thermodynamic analysis. Also, strong conclusions are made with limited analysis to support them. For example, a major finding seems to be that these changes in emissions altered the aerosol formation mechanisms. It is stated in the Abstract and Conclusions that lower SO2 suppressed rapid sulfate formation through heterogeneous reactions. But it is not clear that this is really shown or tested in a quantitative way, instead it seems to be mostly speculation, based mainly on this hypotheses being consistent with the observations. Line 324 simply states: the results here imply that: : : Could not a chemical transport model, or maybe a simpler 0-D or 1D box model be run to further test this hypothesis? Overall, my recommendation is that the authors check closely if their reported findings are truly supported in the manuscript by quantitative analyses, and if only based on a consistency with expectations, that this clearly be stated. More detailed comments are provided below.

We would like to thank Rodney Weber for giving the constructive and helpful comments and suggestions, especially for the discussions on thermodynamic analysis. In the revised manuscript, more technical details about the thermodynamic analysis have been added according to the comments. One finding that the decrease in $SO_2$ emissions suppressed the rapid sulfate formation through heterogeneous reactions was speculated based on the ambient observations. We found that compared to the fast $SO_2$-to-sulfate formation starting from a RH threshold of ~50% in 2014, the promptly increased sulfate formation through heterogeneous reactions was observed to delay to a higher RH of 70% in 2017. Therefore, this is one hypothesis based on the consistency with ambient observations. We have clearly stated it in the Abstract, Sect. 3.2.4, and Conclusions in the manuscript.

In the following, we will answer the comments point by point.

Specific Comments:

Line 59-60; I would think atmospheric chemical reactions (secondary aerosol) and deposition would also be a major contributor to PM2.5 composition.

Agreed. We slightly modified the sentence to "The chemical composition of $PM_{2.5}$ is mainly affected by the following factors: precursor emissions, meteorological conditions, atmospheric chemical reactions, and regional transport and deposition".

Line 136 to 137, in giving the ambient data vs model comparison, state the integration time, ie 24 hr average data?

It has been clearly stated that the data are 24-hour averages.

ISORROPIA calculations: In this paper the model is run without considering nonvolatile cations. Maybe a few words should be added why this is ok, ie, it may be argued that for PM1 this is reasonable. As another example, the nitrate considered in the paper is all semivolatile nitrate (ie, NH4NO3), but it is possible that nonvolatile nitrate also exists in the ambient PM2.5 (eg, Ca(NO3)2). Thus if Ca2+ was considered in the thermodynamic calculations, it could affect predicted pH and NO3- concentrations. Most of this Ca2+ would likely be in the 1 to 2.5 um range, and since the comparison between PM1 and PM2.5 mass is reasonable, the authors could argue that it is not a large contribution. Also, I suggest the authors specifically note what RH range was used in the thermodynamic calculations, many of the assumptions, such as no separate organic/inorganic phases,

etc, may be less likely at lower RH. (say <40 to 50%). Line 231-232 notes that the observed RH was about 33 to 34% in the winter of 2017. This a very low RH to comfortably run ISORROPIA under the metastable assumption without some test on the reliability of the results.

Indeed, including nonvolatile cations in ISORROPIA calculations would influence the model results. But as the reviewer said, nonvolatile cations, i.e., $Na^+$, $K^+$, $Ca^{2+}$, $Mg^{2+}$, mainly exists in the size range of 1.0 to 2.5 µm in particles and has a minor contribution to $PM_1$. Therefore, nonvolatile cations are not considered in pH calculation in this study. A previous study by Song et al. (2018) showed that including nonvolatile cations in ISORROPIA calculations did not significantly change the particle pH. Discussions about the effects of nonvolatile cations on pH calculations have been added in the manuscript "The effects of nonvolatile cations (i.e., $Na^+$, $K^+$, $Ca^{2+}$, $Mg^{2+}$) are not considered in this study because the fraction of nonvolatile cations in $PM_1$ in Beijing is generally negligible compared to $SO_4^{2-}$, $NO_3^-$, and $NH_4^+$ (Sun et al., 2014). Although nonvolatile nitrate may exist in ambient particles as $Ca(NO_3)_2$ and $Mg(NO_3)_2$, $Ca^{2+}$ and $Mg^{2+}$ are mainly abundant at sizes above 1 µm (Zhao et al., 2017). In addition, the mixing state of $PM_1$ nonvolatile cations with $SO_4^{2-}$, $NO_3^-$, and $NH_4^+$ remains to be investigated (Guo et al., 2016, 2017). Previous studies showed that including the nonvolatile cations in ISORROPIA-II does not significantly affect the pH calculations unless the cations become important relative to anions (Guo et al., 2016; Song et al., 2018). The sensitivity test for Beijing winter conditions suggested that with nonvolatile cations, the predicted pH values increase by about 0.1 units."

We agree with the reviewer that RH ranges influence the liquid or solid phases of atmospheric aerosols. So far, there are no observational data showing whether aerosols are in a metastable or stable state in Beijing wintertime (Song et al., 2018). According to previous studies, at low RH, especially when RH < 20% or 30%, aerosols are less likely to be in a completely liquid state (Fountoukis and Nenes, 2007; Guo et al., 2016, 2017). Therefore, we exclude periods when RH < 30% in this study. After that, an average RH value of 50% is now used in the thermodynamic calculations assuming that aerosols were in metastable states. We also did a sensitivity study assuming that solid phases are present. For that case, over 88% of the data resulted in pH values approximating 7.6 with few variations, which is unrealistic. After the correction, the average pH values for year 2013, 2014, and 2017 are 4.5, 4.8, and 5.3, respectively. The results indicate a moderately acidic condition for aerosols in Beijing in winter, consistent with previous studies (Guo et al., 2017; Liu et al., 2017; Song et al., 2018). The correction did not change the trend of pH variation from 2013 to 2017 because the reduced sulfate concentration played a dominant role in pH variation. The corresponding explanations and corrections have been added and updated in the manuscript. "Up to now, there are no observational data showing whether aerosols are in a metastable or stable state in Beijing in winter (Song et al., 2018). According to previous studies, at low RH (RH < 20% or 30%), aerosols are less likely to be in a completely liquid state (Fountoukis and Nenes, 2007; Guo et al., 2016, 2017). Therefore, periods with RH < 30% were excluded in this study." "During 2013-2017, the average particle pH varied from 4.5 to 5.3, with a significant decrease in sulfate concentration, resulting in a more neutral atmospheric environment. The pH values here agree reasonably with previous ISORROPIA-II calculations, showing that fine particles are moderately acidic in northern China during wintertime (Guo et al., 2017; Liu et al., 2017; Song et al., 2018)."

Figures in the manuscript showing the results of thermodynamic analysis have also been updated:

[Figure]

Figure 9. Sensitivity of (a) AWC and (b) particle pH to the mass concentrations of particulate sulfate and nitrate. The stars indicate the average winter conditions for the years 2013, 2014, and 2017.

[Figure]

Figure 10. Sensitivity of the ammonium partitioning ratio to the mass concentrations of particulate sulfate and nitrate. The stars indicate the average winter conditions for the years 2013, 2014, and 2017.

In the experimental section, there was no discussion of measurement of HNO3 or NH3 (gas species), yet ISORROPIA was run such that these data are needed, ie, run in forward mode. More information is needed in the paper on how the model was run without these critical gas phase species.

To investigate how the variations in particulate nitrate and sulfate concentrations affect aerosol properties, this study used ISORROPIA-II to generate the contour plots in Fig. 9 and Fig. 10. ISORROPIA-II was run in the forward mode, which calculates the equilibrium partitioning with the total concentration of both gas and particle phase species. Previous study shows that the forward mode is less sensitive to measurement error than the reverse mode (Hennigan et al., 2015). To run the model, a selected sulfate concentration with the average temperature, RH, and total ammonia concentration ($NH_3 + NH_4^+$) during the winters of 2014 and 2017 was input to ISORROPIA-II, where the total nitrate concentration ($HNO_3 + NO_3^-$) was left as the free variable. The gaseous $HNO_3$ and $NH_3$ concentrations were not directly measured in this work. To estimate the $NH_3$ concentration, an empirical equation derived based on long-term measurements in winter in Beijing was applied, $NH_3$ (ppb) = 0.34 × $NO_x$ (ppb) + 0.63 (Meng et al., 2011). On average, the $NH_3$ concentration was

estimated to be around 14.0 μg m$^{-3}$ during the winters of 2014 and 2017 in Beijing, consistent with previous observations in the same season (Meng et al., 2011; Zhao et al., 2016; Zhang et al., 2018). For gaseous HNO$_3$, the total NO$_3^-$ concentration (HNO$_3$+aerosol NO$_3^-$) varying from 0.2 to 75 μg m$^{-3}$ was used as the input.

More discussions about the consideration of gaseous HNO$_3$ and NH$_3$ concentrations have been added in the manuscript as follows: "The gaseous HNO$_3$ and NH$_3$ concentrations were not directly measured during our campaign. But long-term measurements in Beijing showed that gaseous NH$_3$ concentration correlated well with NO$_x$ concentration in winter (Meng et al., 2011). Therefore, the empirical equation derived from Meng et al. (2011), NH$_3$ (ppb) = 0.34 × NO$_x$ (ppb) + 0.63, was applied to estimate the gaseous NH$_3$ concentration. On average, the NH$_3$ concentration was approximated to be 14.0 μg m$^{-3}$ during the winters of 2014 and 2017, consistent with previous observations jn the same season of Beijing (Meng et al., 2011; Zhao et al., 2016; Zhang et al., 2018). The total nitrate concentration, including both gaseous HNO$_3$ and particulate nitrate, varied from 0.2 to 75 μg m$^{-3}$ for the sensitivity study."

Line 175 or in Table 1 title state this is observational data (not model).

It has been clearly stated in the title of Table 1.

Line 258, what does Until 2017 mean? These changes were completed by 2017?

It means that these changes have been completed by the end of 2017. To make it more clearly, we changed "Until 2017" to "By the end of 2017".

Line 311, does not tell the complete story. There are publications, see below, that show the predictions of Cheng et al. 2016 and Wang et al, 2016 are likely not correct due to their incorrect calculation of fine particle pH and that this proposed heterogeneous chemistry is highly sensitive to pH. This counter argument should also be noted here in this paper for completeness.

Liu, M., Y. Song, T. Zhoh, Z. Xu, C. Yan, M. Zheng, Z. Wu, M. Hu, Y. Wu, and T. Zhu (2017), Fine Particle pH during Severe Haze Episodes in Northern China, Geophys. Res. Lett., 10.1002/2017GL073210.

Guo, H., R. J. Weber, and A. Nenes (2017), High levels of ammonia do not raise fine particle pH sufficiently to yield nitrogen oxide-dominated sulfate production, Sci. Reports, 7(12109), DOI:10.1038/s41598-41017-11704-41590.

Song, S., M. Gao, W. Xu, J. Shao, G. Shi, S. Wang, Y. Wang, Y. Sun, and M. B. McElroy (2018), Fine particle pH for Beijing winter haze as inferred from different thermodynamic equilibrium models, Atm. Chem. Phys., 18, 7423-7438.

Agreed. The corresponding argument has been noted in the manuscript as follows: "Recently, studies have found that SO$_2$ oxidation by NO$_2$ in aerosol water with near neutral aerosol acidity, which is usually ignored in current model simulations, plays an important role in the persistent formation of sulfate during haze events in northern China (B. Zheng et al., 2015; Cheng et al., 2016; Wang et al., 2016). Others pointed out that regardless of the high NH$_3$ levels, aerosols are always moderately acidic in northern China, and there are probably other alternative formation pathways contributing to fast sulfate production in haze pollution (Guo et al., 2017b; Liu et al., 2017; Song et al., 2018)."

Line 318-319, define SOR and NOR (ie, what does the acronym stand for?)

The SOR and NOR stand for sulfur oxidation ratio and nitrogen oxidation ratio, respectively. This has been clearly clarified in the manuscript as "The sulfur oxidation ratio (SOR) and nitrogen oxidation ratio (NOR) were further estimated as the molar ratio of sulfate to the sum of sulfate and SO$_2$ and the molar ratio of nitrate to the sum of nitrate and NO$_x$, respectively, to quantify the degree of SO$_2$ and NO$_x$ oxidations (Zheng et al., 2015; Li et al., 2016)."

Lines 339 to 344. Although this discussion tends to follow the Seinfeld and Pandis discussion of sulfate/nitrate/ammonium interactions, it is largely based on weak intuitive arguments and not a rigorous thermodynamic discussion. It is suggested that these types of statements be avoided. Below are possible other ways to discuss the interplay between sulfate, nitrate and ammonium and gas species, nitric acid and ammonia:

Change: Particulate nitrate in PM2.5 is mainly formed through the neutralization of HNO3 with NH3 to something like: Semivolatile PM2.5 particulate nitrate is formed through the partitioning of HNO3 to the particle phase, which is more favored at higher aerosol pH. pH is affected by gas phase NH3 concentrations, where higher NH3 generally leads to higher pH and so possibly more particulate nitrate.

Thanks for the suggestions. We have revised the text accordingly.

Change: Nitrate formation was also affected by the competition for available NH3 between sulfate and nitrate. In the atmosphere, NH3 prefers to react first with H2SO4 to form ammonium sulfate due its stability. To: Because sulfate is nonvolatile, when it is a significant fraction of the aerosol mass it has a dominant influence on aerosol pH, making the aerosol acidic (low pH). In contrast ammonium and nitrate are semivolatile and so their particle-phase concentrations depend on the meteorological conditions (T, RH) their corresponding gas phase concentrations, (NH3 and HNO3 respectively) and aerosol pH. For example, at high sulfate and moderate NH3 concentrations the aerosol can be too acidic for partitioning of HNO3, but at higher NH3, or if sulfate concentrations drop sufficiently (or RH increases), particle pH will increase and can reach a point at which HNO3 partitioning can occur and nitrate aerosol formed. Lower T also favors partitioning to the particle phase through Henry's law constants.

Thanks for the suggestions. We have modified the text accordingly.

As for the last line, when RH>60% maybe an additional explanation for the trend in Fig8d is that as RH increases liquid water levels increase resulting in higher pH, which allows more nitrate to partition to the aerosol through a feedback loop, as is discussed later in the paper. That is, the increase in NOR may be due to more than just heterogeneous nitrate production.

Agreed. This additional explanation has been added as "In addition, as RH increases, the AWC increases accordingly, resulting in higher aerosol pH. This allows more semivolatile nitrate to partition to the particle phase through a feedback loop, thus favoring the formation of particulate nitrate."

Line 366-369, In Fig S9a provide a reference for the calculation of epsilon(NO3-). This pH of 3 at which the sensitivity of epsilon(NO3-) changes, as found in this work, was discussed in detail by Guo et al., ACP 2018, which should be cited.

A reference for the calculation of $\in (NO_3^-)$ has been added. The reference of Guo et al., ACP 2018, has been cited.

Finally, there are a number of studies reporting pH in Beijing over different periods. Some of these did analysis to test the thermodynamic model predictions, which was not done here. A summary of these papers and comparison to pH reported in this paper is warranted to support this analysis.

According to the comments above regarding the thermodynamic analysis, summarization of previous studies reporting pH in Beijing and comparison with results in this study have been included in the manuscript. In addition, more information were added: "Previous studies showed that including the nonvolatile cations in ISORROPIA-II does not significantly affect the pH calculations unless the cations become important relative to anions (Guo et al., 2016; Song et al., 2018). The sensitivity test for Beijing winter conditions suggested that with nonvolatile cations, the predicted pH values increase by about 0.1 units."

**References**

Fountoukis, C., and Nenes, A.: ISORROPIA II: a computationally efficient thermodynamic equilibrium model for K+-Ca2+-Mg2+-NH4+-Na+-SO42--NO3--Cl--H2O aerosols, Atmos Chem Phys, 7, 4639-4659, 2007.

Guo, H., Sullivan, A. P., Campuzano-Jost, P., Schroder, J. C., Lopez-Hilfiker, F. D., Dibb, J. E., Jimenez, J. L., Thornton, J. A., Brown, S. S., Nenes, A., and Weber, R. J.: Fine particle pH and the partitioning of nitric acid during winter in the northeastern United States, Journal of Geophysical Research: Atmospheres, 121, 10,355-310,376, 10.1002/2016JD025311, 2016.

Guo, H., Weber, R. J., and Nenes, A.: High levels of ammonia do not raise fine particle pH sufficiently to yield nitrogen oxide-dominated sulfate production, Scientific Reports, 7, 12109, 10.1038/s41598-017-11704-0, 2017.

Hennigan, C. J., Izumi, J., Sullivan, A. P., Weber, R. J., and Nenes, A.: A critical evaluation of proxy methods used to estimate the acidity of atmospheric particles, Atmos. Chem. Phys., 15, 2775-2790, https://doi.org/10.5194/acp-15-2775-2015, 2015.

Liu, M., Song, Y., Zhou, T., Xu, Z., Yan, C., Zheng, M., Wu, Z., Hu, M., Wu, Y., and Zhu, T.: Fine particle pH during severe haze episodes in northern China, Geophys Res Lett, 44, 5213-5221, 10.1002/2017GL073210, 2017.

Meng, Z. Y., Lin, W. L., Jiang, X. M., Yan, P., Wang, Y., Zhang, Y. M., Jia, X. F., and Yu, X. L.: Characteristics of atmospheric ammonia over Beijing, China, Atmos. Chem. Phys., 11, 6139-6151, https://doi.org/10.5194/acp-11-6139-2011, 2011.

Song, S., Gao, M., Xu, W., Shao, J., Shi, G., Wang, S., Wang, Y., Sun, Y., and McElroy, M. B.: Fine-particle pH for Beijing winter haze as inferred from different thermodynamic equilibrium models, Atmos. Chem. Phys., 18, 7423-7438, https://doi.org/10.5194/acp-18-7423-2018, 2018.

Zhao, M., Wang, S., Tan, J., Hua, Y., Wu, D., and Hao, J.: Variation of Urban Atmospheric Ammonia Pollution and its Relation with PM2.5 Chemical Property in Winter of Beijing, China, Aerosol Air Qual Res, 16, 1378-1389, 10.4209/aaqr.2015.12.0699, 2016.

Zhang, Y., Tang, A., Wang, D., Wang, Q., Benedict, K., Zhang, L., Liu, D., Li, Y., Collett Jr., J. L., Sun, Y., and Liu, X.: The vertical variability of ammonia in urban Beijing, China, Atmos. Chem. Phys., 18, 16385-16398, https://doi.org/10.5194/acp-18-16385-2018, 2018.

---

## Author Comment (AC2) · 29 Jul 2019

The comment was uploaded in the form of a supplement:
https://www.atmos-chem-phys-discuss.net/acp-2019-450/acp-2019-450-AC2-supplement.pdf

————————————————————

---

## Author Response (AR1)

**Response to Referee Comment 1 (RC1) on "Rapid transition in winter aerosol composition in Beijing from 2014 to 2017: response to clean air actions" by H. Li et al.**

In past publications, the authors have reported on the effects of emission controls on China's air quality. This paper focuses on how those emission changes affect aerosol chemical composition. The authors present a comprehensive analysis involving measurements and models to assess the role of just emissions, apart from transport and meteorological differences. The results are interesting and the topic appropriate for this journal, however some details were missing and in many cases the analysis or description of processes was not of sufficient technical detail, such as the thermodynamic analysis. Also, strong conclusions are made with limited analysis to support them. For example, a major finding seems to be that these changes in emissions altered the aerosol formation mechanisms. It is stated in the Abstract and Conclusions that lower SO2 suppressed rapid sulfate formation through heterogeneous reactions. But it is not clear that this is really shown or tested in a quantitative way, instead it seems to be mostly speculation, based mainly on this hypotheses being consistent with the observations. Line 324 simply states: the results here imply that: : : Could not a chemical transport model, or maybe a simpler 0-D or 1D box model be run to further test this hypothesis? Overall, my recommendation is that the authors check closely if their reported findings are truly supported in the manuscript by quantitative analyses, and if only based on a consistency with expectations, that this clearly be stated. More detailed comments are provided below.

We would like to thank Rodney Weber for giving the constructive and helpful comments and suggestions, especially for the discussions on thermodynamic analysis. In the revised manuscript, more technical details about the thermodynamic analysis have been added according to the comments. One finding that the decrease in  $SO_2$  emissions suppressed the rapid sulfate formation through heterogeneous reactions was speculated based on the ambient observations. We found that compared to the fast  $SO_2$ -to-sulfate formation starting from a RH threshold of ~50% in 2014, the promptly increased sulfate formation through heterogeneous reactions was observed to delay to a higher RH of 70% in 2017. Therefore, this is one hypothesis based on the consistency with ambient observations. We have clearly stated it in the Abstract, Sect. 3.2.4, and Conclusions in the manuscript.

In the following, we will answer the comments point by point.

Specific Comments:

Line 59-60; I would think atmospheric chemical reactions (secondary aerosol) and deposition would also be a major contributor to PM2.5 composition.

Agreed. We slightly modified the sentence to "The chemical composition of  $PM_{2.5}$  is mainly affected by the following factors: precursor emissions, meteorological conditions, atmospheric chemical reactions, and regional transport and deposition".

Line 136 to 137, in giving the ambient data vs model comparison, state the integration time, ie 24 hr average data?

It has been clearly stated that the data are 24-hour averages.

ISORROPIA calculations: In this paper the model is run without considering nonvolatile cations. Maybe a few words should be added why this is ok, ie, it may be argued that for PM1 this is reasonable. As another example, the nitrate considered in the paper is all semivolatile nitrate (ie, NH4NO3), but it is possible that nonvolatile nitrate also exists in the ambient PM2.5 (eg, Ca(NO3)2). Thus if Ca2+ was considered in the thermodynamic calculations, it could affect predicted pH and NO3- concentrations. Most of this Ca2+ would likely be in the 1 to 2.5 um range, and since the comparison between PM1 and PM2.5 mass is reasonable, the authors could argue that it is not a large contribution. Also, I suggest the authors specifically note what RH range was used in the thermodynamic calculations, many of the assumptions, such as no separate organic/inorganic phases,

etc, may be less likely at lower RH. (say <40 to 50%). Line 231-232 notes that the observed RH was about 33 to 34% in the winter of 2017. This a very low RH to comfortably run ISORROPIA under the metastable assumption without some test on the reliability of the results.

Indeed, including nonvolatile cations in ISORROPIA calculations would influence the model results. But as the reviewer said, nonvolatile cations, i.e., Na+, K+, Ca2+, Mg2+, mainly exists in the size range of 1.0 to 2.5  $\mu$ m in particles and has a minor contribution to PM1. Therefore, nonvolatile cations are not considered in pH calculation in this study. A previous study by Song et al. (2018) showed that including nonvolatile cations in ISORROPIA calculations did not significantly change the particle pH. Discussions about the effects of nonvolatile cations on pH calculations have been added in the manuscript "The effects of nonvolatile cations in PM1 in Beijing is generally negligible compared to SO42-, NO3-, and NH4+ (Sun et al., 2014). Although nonvolatile nitrate may exist in ambient particles as Ca(NO3)2 and Mg(NO3)2, Ca2+ and Mg2+ are mainly abundant at sizes above 1  $\mu$ m (Zhao et al., 2017). In addition, the mixing state of PM1 nonvolatile cations with SO42-, NO3-, and NH4+ remains to be investigated (Guo et al., 2016, 2017). Previous studies showed that including the nonvolatile cations in ISORROPIA-II does not significantly affect the pH calculations unless the cations become important relative to anions (Guo et al., 2016; Song et al., 2018). The sensitivity test for Beijing winter conditions suggested that with nonvolatile cations, the predicted pH values increase by about 0.1 units."

We agree with the reviewer that RH ranges influence the liquid or solid phases of atmospheric aerosols. So far, there are no observational data showing whether aerosols are in a metastable or stable state in Beijing wintertime (Song et al., 2018). According to previous studies, at low RH, especially when RH

Figure 9. Sensitivity of (a) AWC and (b) particle pH to the mass concentrations of particulate sulfate and nitrate. The stars indicate the average winter conditions for the years 2013, 2014, and 2017.

Figure 10. Sensitivity of the ammonium partitioning ratio to the mass concentrations of particulate sulfate and nitrate. The stars indicate the average winter conditions for the years 2013, 2014, and 2017.

In the experimental section, there was no discussion of measurement of HNO3 or NH3 (gas species), yet ISORROPIA was run such that these data are needed, ie, run in forward mode. More information is needed in the paper on how the model was run without these critical gas phase species.

To investigate how the variations in particulate nitrate and sulfate concentrations affect aerosol properties, this study used ISORROPIA-II to generate the contour plots in Fig. 9 and Fig. 10. ISORROPIA-II was run in the forward mode, which calculates the equilibrium partitioning with the total concentration of both gas and particle phase species. Previous study shows that the forward mode is less sensitive to measurement error than the reverse mode (Hennigan et al., 2015). To run the model, a selected sulfate concentration with the average temperature, RH, and total ammonia concentration ( $NH_3 + NH_4^+$ ) during the winters of 2014 and 2017 was input to ISORROPIA-II, where the total nitrate concentration ( $HNO_3 + NO_3^-$ ) was left as the free variable. The gaseous HNO3 and NH3 concentrations were not directly measured in this work. To estimate the NH3 concentration, an empirical equation derived based on long-term measurements in winter in Beijing was

applied, NH3 (ppb) =  $0.34 \times NO_x$  (ppb) + 0.63 (Meng et al., 2011). On average, the NH3 concentration was estimated to be around 14.0 µg m-3 during the winters of 2014 and 2017 in Beijing, consistent with previous observations in the same season (Meng et al., 2011; Zhao et al., 2016; Zhang et al., 2018). For gaseous HNO3, the total NO3- concentration (HNO3+aerosol NO3-) varying from 0.2 to 75 µg m-3 was used as the input.

More discussions about the consideration of gaseous HNO3 and NH3 concentrations have been added in the manuscript as follows: "The gaseous HNO3 and NH3 concentrations were not directly measured during our campaign. But long-term measurements in Beijing showed that gaseous NH3 concentration correlated well with NOx concentration in winter (Meng et al., 2011). Therefore, the empirical equation derived from Meng et al. (2011), NH3 (ppb) =  $0.34 \times NO_x$  (ppb) + 0.63, was applied to estimate the gaseous NH3 concentration. On average, the NH3 concentration was approximated to be 14.0 µg m-3 during the winters of 2014 and 2017, consistent with previous observations jn the same season of Beijing (Meng et al., 2011; Zhao et al., 2016; Zhang et al., 2018). The total nitrate concentration, including both gaseous HNO3 and particulate nitrate, varied from 0.2 to 75 µg m-3 for the sensitivity study."

Line 175 or in Table 1 title state this is observational data (not model).

It has been clearly stated in the title of Table 1.

Line 258, what does Until 2017 mean? These changes were completed by 2017?

It means that these changes have been completed by the end of 2017. To make it more clearly, we changed "Until 2017" to "By the end of 2017".

Line 311, does not tell the complete story. There are publications, see below, that show the predictions of Cheng et al. 2016 and Wang et al, 2016 are likely not correct due to their incorrect calculation of fine particle pH and that this proposed heterogeneous chemistry is highly sensitive to pH. This counter argument should also be noted here in this paper for completeness.

Liu, M., Y. Song, T. Zhoh, Z. Xu, C. Yan, M. Zheng, Z. Wu, M. Hu, Y. Wu, and T. Zhu (2017), Fine Particle pH during Severe Haze Episodes in Northern China, Geophys. Res. Lett., 10.1002/2017GL073210.

Guo, H., R. J. Weber, and A. Nenes (2017), High levels of ammonia do not raise fine particle pH sufficiently to yield nitrogen oxide-dominated sulfate production, Sci. Reports, 7(12109), DOI:10.1038/s41598-41017-11704-41590.

Song, S., M. Gao, W. Xu, J. Shao, G. Shi, S. Wang, Y. Wang, Y. Sun, and M. B. McElroy (2018), Fine particle pH for Beijing winter haze as inferred from different thermodynamic equilibrium models, Atm. Chem. Phys., 18, 7423-7438.

Agreed. The corresponding argument has been noted in the manuscript as follows: "Recently, studies have found that  $SO_2$  oxidation by  $NO_2$  in aerosol water with near neutral aerosol acidity, which is usually ignored in current model simulations, plays an important role in the persistent formation of sulfate during haze events in northern China (B. Zheng et al., 2015; Cheng et al., 2016; Wang et al., 2016). Others pointed out that regardless of the high NH3 levels, aerosols are always moderately acidic in northern China, and there are probably other alternative formation pathways contributing to fast sulfate production in haze pollution (Guo et al., 2017; Liu et al., 2017; Song et al., 2018)."

**Line 318-319, define SOR and NOR (ie, what does the acronym stand for?)**

The SOR and NOR stand for sulfur oxidation ratio and nitrogen oxidation ratio, respectively. This has been clearly clarified in the manuscript as "The sulfur oxidation ratio (SOR) and nitrogen oxidation ratio (NOR) were further estimated as the molar ratio of sulfate to the sum of sulfate and SO2 and the molar ratio of nitrate to the sum of nitrate and NOx, respectively, to quantify the degree of SO2 and NOx oxidations (Zheng et al., 2015; Li et al., 2016)."

Lines 339 to 344. Although this discussion tends to follow the Seinfeld and Pandis discussion of sulfate/nitrate/ammonium interactions, it is largely based on weak intuitive arguments and not a rigorous thermodynamic discussion. It is suggested that these types of statements be avoided. Below are possible other ways to discuss the interplay between sulfate, nitrate and ammonium and gas species, nitric acid and ammonia:

Change: Particulate nitrate in PM2.5 is mainly formed through the neutralization of HNO3 with NH3 to something like: Semivolatile PM2.5 particulate nitrate is formed through the partitioning of HNO3 to the particle phase, which is more favored at higher aerosol pH. pH is affected by gas phase NH3 concentrations, where higher NH3 generally leads to higher pH and so possibly more particulate nitrate.

**Thanks for the suggestions. We have revised the text accordingly.**

Change: Nitrate formation was also affected by the competition for available NH3 between sulfate and nitrate. In the atmosphere, NH3 prefers to react first with H2SO4 to form ammonium sulfate due its stability. To: Because sulfate is nonvolatile, when it is a significant fraction of the aerosol mass it has a dominant influence on aerosol pH, making the aerosol acidic (low pH). In contrast ammonium and nitrate are semivolatile and so their particle-phase concentrations depend on the meteorological conditions (T, RH) their corresponding gas phase concentrations, (NH3 and HNO3 respectively) and aerosol pH. For example, at high sulfate and moderate NH3 concentrations the aerosol can be too acidic for partitioning of HNO3, but at higher NH3, or if sulfate concentrations drop sufficiently (or RH increases), particle pH will increase and can reach a point at which HNO3 partitioning can occur and nitrate aerosol formed. Lower T also favors partitioning to the particle phase through Henry's law constants.

**Thanks for the suggestions. We have modified the text accordingly.**

As for the last line, when RH>60% maybe an additional explanation for the trend in Fig8d is that as RH increases liquid water levels increase resulting in higher pH, which allows more nitrate to partition to the aerosol through a feedback loop, as is discussed later in the paper. That is, the increase in NOR may be due to more than just heterogeneous nitrate production.

Agreed. This additional explanation has been added as "In addition, as RH increases, the AWC increases accordingly, resulting in higher aerosol pH. This allows more semivolatile nitrate to particle phase through a feedback loop, thus favoring the formation of particulate nitrate."

Line 366-369, In Fig S9a provide a reference for the calculation of epsilon(NO3-). This pH of 3 at which the sensitivity of epsilon(NO3-) changes, as found in this work, was discussed in detail by Guo et al., ACP 2018, which should be cited.

**A reference for the calculation of $\in (NO_3^-)$ has been added. The reference of Guo et al., ACP 2018, has been cited.**

Finally, there are a number of studies reporting pH in Beijing over different periods. Some of these did analysis to test the thermodynamic model predictions, which was not done here. A summary of these papers and comparison to pH reported in this paper is warranted to support this analysis.

According to the comments above regarding the thermodynamic analysis, summarization of previous studies reporting pH in Beijing and comparison with results in this study have been included in the manuscript. In addition, more information were added: "Previous studies showed that including the nonvolatile cations in ISORROPIA-II does not significantly affect the pH calculations unless the cations become important relative to anions (Guo et al., 2016; Song et al., 2018). The sensitivity test for Beijing winter conditions suggested that with nonvolatile cations, the predicted pH values increase by about 0.1 units."

References

Fountoukis, C., and Nenes, A.: ISORROPIA II: a computationally efficient thermodynamic equilibrium model for K+-Ca2+-Mg2+-NH4+-Na+-SO42--NO3--Cl--H2O aerosols, Atmos Chem Phys, 7, 4639-4659, 2007.

Guo, H., Sullivan, A. P., Campuzano-Jost, P., Schroder, J. C., Lopez-Hilfiker, F. D., Dibb, J. E., Jimenez, J. L., Thornton, J. A., Brown, S. S., Nenes, A., and Weber, R. J.: Fine particle pH and the partitioning of nitric acid during winter in the northeastern United States, Journal of Geophysical Research: Atmospheres, 121, 10,355-310,376, 10.1002/2016JD025311, 2016.

Guo, H., Weber, R. J., and Nenes, A.: High levels of ammonia do not raise fine particle pH sufficiently to yield nitrogen oxide-dominated sulfate production, Scientific Reports, 7, 12109, 10.1038/s41598-017-11704-0, 2017.

Hennigan, C. J., Izumi, J., Sullivan, A. P., Weber, R. J., and Nenes, A.: A critical evaluation of proxy methods used to estimate the acidity of atmospheric particles, Atmos. Chem. Phys., 15, 2775-2790, https://doi.org/10.5194/acp-15-2775-2015, 2015.

Liu, M., Song, Y., Zhou, T., Xu, Z., Yan, C., Zheng, M., Wu, Z., Hu, M., Wu, Y., and Zhu, T.: Fine particle pH during severe haze episodes in northern China, Geophys Res Lett, 44, 5213-5221, 10.1002/2017GL073210, 2017.

Meng, Z. Y., Lin, W. L., Jiang, X. M., Yan, P., Wang, Y., Zhang, Y. M., Jia, X. F., and Yu, X. L.: Characteristics of atmospheric ammonia over Beijing, China, Atmos. Chem. Phys., 11, 6139-6151, https://doi.org/10.5194/acp-11-6139-2011, 2011.

Song, S., Gao, M., Xu, W., Shao, J., Shi, G., Wang, S., Wang, Y., Sun, Y., and McElroy, M. B.: Fine-particle pH for Beijing winter haze as inferred from different thermodynamic equilibrium models, Atmos. Chem. Phys., 18, 7423-7438, https://doi.org/10.5194/acp-18-7423-2018, 2018.

Zhao, M., Wang, S., Tan, J., Hua, Y., Wu, D., and Hao, J.: Variation of Urban Atmospheric Ammonia Pollution and its Relation with PM2.5 Chemical Property in Winter of Beijing, China, Aerosol Air Qual Res, 16, 1378-1389, 10.4209/aaqr.2015.12.0699, 2016.

Zhang, Y., Tang, A., Wang, D., Wang, Q., Benedict, K., Zhang, L., Liu, D., Li, Y., Collett Jr., J. L., Sun, Y., and Liu, X.: The vertical variability of ammonia in urban Beijing, China, Atmos. Chem. Phys., 18, 16385-16398, https://doi.org/10.5194/acp-18-16385-2018, 2018.

**Response to Referee Comment 2 (RC2) on "Rapid transition in winter aerosol composition in Beijing from 2014 to 2017: response to clean air actions" by H. Li et al.**

This study combines in-situ measurements of PM1 composition in Beijing during the winters of 2014 and 2017 and the simulation results from a regional chemical transport model to investigate the impacts that the clean air actions in China have had on aerosol chemistry. The relative contributions of anthropogenic emissions, meteorological conditions, and regional transport to the changes in aerosol composition in Beijing are also investigated. This is an interesting work and provides a timely and relevant analysis of a current problem – how aerosol pollution in Beijing responded to the implementation of the Air Pollution Prevent and Control Policy. Overall, the manuscript is well written and fits well within ACP's aims and scope. However, the current version may require substantial revision before publication can be considered. A major shortcoming in this manuscript is that the Methods section is very much lacking of important technical details, for example on aerosol source contribution analysis and CMT model performance, thereby raises doubts about the credibility of the results. Moreover, there are some inconsistencies in the results and discussions which could cause concerns about the quality of the data, representativeness of the findings, or validity of the conclusions. Additionally, the figure legends and captions are often too brief to make the figures understandable.

We thank the reviewer for the positive feedback and helpful comments. In the revised manuscript, more technical details in the Method section have been added. In the following, we will answer the comments point by point.

**Detailed comments:**

Line 44-46, how much did air pollutants reduce between 2013 – 2017 in the Beijing area?

Emission changes of different air pollutants in Beijing with the implementation of the clean air actions was analyzed in details in Section 3.2.2 and Figure 6.

Section 2.2. is cursory and provides very little information on the organic aerosol source apportionment analysis. Details must be provided on how the PMF/ME-2 analysis of the ACSM data was performed, what data treatments were implemented, and how the solution conditions (eg number of factors, a value, fpeak) were selected and evaluated.

For the ME-2 analysis of the ACSM organic data, the mass spectra and error matrices were prepared based on the procedures given by Ulbrich et al. (2009) and Zhang et al. (2011). Detailed evaluations of different solutions were provided in the Supplement (Figs. S2-11), including the factor time series, mass spectra, and diurnal patterns with different *a* values. Corresponding changes can be found in the revised manuscript and Supplement.

In Section 2.3. more information should be provided on the performance of the CMT model at simulating PM2.5 composition and how the modeling results compare to observations for 2014 and 2017 winters separately.

**As the reviewer suggested, we added more information about the evaluation of the CMT model performance in Section 2.3 and the Supplement (Table S1-2, Fig. S12).**

Section 2.5 lists three assumptions about aerosol properties in the ISORROPIA modeling. Several references are cited and claimed to support these assumptions for this study. But upon reading the references more closely, they don't seem so as the references either talked about aerosols from different locations or under different meteorological conditions, or simply did not provide direct evidence on aerosol physical states. In fact, given the wintry weather (very low RH and T) and intense local emissions in Beijing, it is hard to believe that internal mixing and single aqueous phase were the prevailing aerosol conditions relevant to this work.

When running the ISORROPIA model, it is assumed that aerosols are internally mixed and composed of a single aqueous phase. Previous study in Beijing during wintertime showed that ISORROPIA predictions with these assumptions were in good agreement with observations (Liu et al., 2017). Up to now, there are no observational data showing whether aerosols are in a metastable (only liquid) or stable (solid plus liquid) state in Beijing in winter (Song et al., 2018). At low RH (RH < 20% or 30%), aerosols are less likely to be in a completely liquid state (Fountoukis and Nenes, 2007; Guo et al., 2016, 2017). Therefore, periods with RH < 30% were excluded in this study and aerosol solutions were assumed to be metastable. Particle liquid-phase separations can occur between the inorganic and organic components. It remains unclear how the phase separations influence pH values. A recent laboratory study suggested that the pH value of the organic-rich fraction under phase separation is about 0.4 units higher than that for a fully mixed aqueous phase (Dallemagne et al., 2016; Song et al., 2018).

**Line 168 – 171, there was no mentioning of NH3 and HNO3 measurements in 2.1., but it is mentioned here that the values of NH3+NH4 and NO3+HNO3 were input into the model. Where did the NH3 and HNO3 data come from?**

In the revised manuscript, discussions were added about how the gas-phase NH3 and HNO3 were considered in this study. "The gaseous HNO3 and NH3 concentrations were not directly measured during our campaign. But long-term measurements in Beijing showed that gaseous NH3 concentration correlated well with NOx concentration in winter (Meng et al., 2011). Therefore, the empirical equation derived from Meng et al. (2011), NH3 (ppb) =  $0.34 \times NO_x$  (ppb) + 0.63, was applied to estimate the gaseous NH3 concentration. On average, the NH3 concentration was approximated to be 14.0 µg m-3 during the winters of 2014 and 2017, consistent with previous observations jn the same season of Beijing (Meng et al., 2011; Zhao et al., 2016; Zhang et al., 2018). The total nitrate concentration, including both gaseous HNO3 and particulate nitrate, varied from 0.2 to 75 µg m-3 for the sensitivity study."

**Line 110, missing numbers after "0." What "a" values were used?**

For the ME-2 analysis, an optimal solution of four factors with the *a* value of 0 was accepted.

**Line 167 -168, what does "the transition in aerosol composition" mean in this sentence?**

The transition in aerosol composition refers to the variations in nitrate and sulfate concentrations. During the winters of 2014-2017, the mass concentrations of both sulfate and nitrate decreased due to the implementation of the clean air actions. The reduction in sulfate concentration was higher than that in nitrate concentration. In this study, we investigated how the variations in nitrate and sulfate concentrations influence particle properties.

Line 186 - 188, the Xu et al. study was also conducted in Beijing in the winters of similar years, but the nitrate to sulfate ratios reported there were much lower than in this study. Normal measurement uncertainty could not explain such large discrepancies (more than a factor of two in difference). Was it due to measurement artifacts or does it suggest some issues with the representativeness of the measurement data? What's the implication for the validity of the conclusions presented in this paper?

The study by Xu et al. (2019b) performed aerosol measurements in Beijing from the mid of November to the mid of December in 2014 and 2016. In this work, field measurements were conducted in Beijing from 6 December to 27 February in the year of 2014 and from 11 December to 2 February in the year of 2017. For the same year of 2014, ambient observations were performed in different months in the study by Xu et al. (2019b) and in this work. Different emission intensities and different meteorological conditions influence the formation of particulate nitrate and sulfate in different months. In this work, one of the conclusions is that the ratio of nitrate/sulfate was higher in winter 2017 than in winter 2014 with the implementation of the clean air actions. The decrease in nitrate concentration was larger in the winter of 2017 than in the winter of 2014. This is consistent with the conclusions by Xu et al. (2019b).

Line 205 - 206, organics were higher than sulfate and nitrate and in both 2014 and 2017, so calling Beijing aerosol pollution being "sulfate-driven" or "nitrate-driven" does not seem logical.

The driving factors in this work refers to the species whose mass fractions increased with the increase of aerosol loadings, consistent with the discussions in previous studies (Wang et al., 2014, Li et al., 2017, 2018; Sun et al., 2018; Xu et al., 2019a). The driving factors favored the development of haze pollution and the increase of aerosol mass concentrations. While organics composed a high fraction of aerosols in both 2014 and 2017, the mass fraction of organics in aerosols decreased with the increase of aerosol loadings.

Line 259, for coal usage reduction in addition to quoting the absolute amount, it would be also interesting in knowing the relative amount of reduction.

The total coal usage for coal-fired boilers is around 20.4 million tons in Beijing in 2013. By the end of 2017, the coal use was reduced by more than 17 million tons in Beijing due to emission controls of coal-fired boilers. In other way, the coal usage in Beijing was reduced by ~83% for coal-fired boilers compared to 2013.

Figure 2, according to the diurnal profiles, HOA concentration in 2014 was 2-3 times lower than 2017. If HOA is representative of emissions from transportation, is this level of decrease consistent with the decrease in emissions according to emission inventory? Moreover, the decrease of BC concentration from 2014 to 2017 was between 30-40% but the reduction of total combustion POA (sum of CCOA, BBOA and HOA) was close to 70%. This would suggest some very large, probably unrealistic, changes in the combustion emission factors.

Compared with 2014, the HOA concentration in 2017 was reduced by around 50%. Consistently, emissions of primary organic carbon from transportation decreased by around 30% in Beijing from 2014 to 2017 according to the emission inventory. Compared to 2014, both BC concentration and POA concentration were reduced largely in 2017. This is not only caused by the changes in the combustion emission factors via the more advanced control technology but also contributed by the reduced usage of traditional fuels, i.e., coal and biomass, through the application of clean energy.

Line 289 – 292, the comparisons of PM1 concentrations between different air trajectory classes do not logically lead to a conclusion about how much Beijing aerosol was influenced by polluted air masses transported from surrounding areas. Beijing has local pollution sources which could cause high PM events as well. In fact, in the paragraph immediately beneath, the authors reported that CMT simulation indicates that regional transport contributes to only 30 -40% of PM2.5 in Beijing.

Back trajectory analysis shows the air masses paths as they move through time and space before they arrive at the receptor site. Different aerosol concentrations for different air masses indicate the influence of regional transport. For example, due to the high emissions of air pollutants and sever aerosol pollutions in the southern surrounding areas of Beijing, air masses transported from the south of Beijing usually showed higher  $PM_1$  concentrations than the air masses from others regions. According to back trajectory analysis, Beijing was less influenced by polluted air masses from surrounding areas in 2017. This is consistent with the results of CMT model simulations, which showed that the lower  $PM_{2.5}$  concentration from regional transport contributed around 40% to  $PM_{2.5}$  reduction in Beijing in 2017. The results are not contradictory to the main conclusion that emission reductions in Beijing and its surrounding regions played a dominant role in air quality improvement during 2014-2017.

Section 3.2.4, 2nd paragraph, what are the rationales for using SO4/BC and NO3/BC ratios in the analysis? Sulfate is a secondary species with formation time scales usually much longer than the emission time scales of BC. So the physical meaning of SO4/BC ratio is unclear.

The absolute concentration of sulfate and nitrate in the atmosphere is not only controlled by atmospheric chemical reactions but also influenced by boundary layer developments. By using SO4/BC and NO3/BC in the analysis, we can see more clearly how the secondary formation of sulfate and nitrate varied compared to primary emissions.

Section 3.2.4, 2nd paragraph, nitate/(nitrate+NOx) is not a proper index for the oxidation ratio of nitrogen. Discussions related to NOR should be either removed or revised.

According to the ISORROPIA calculations in this study, the fraction of particulate nitrate in total nitrate was higher than 0.99 for the average winter conditions in both 2014 and 2017. Therefore, it is meaningful to use nitrate/(nitrate+ $NO_x$ ) to evaluate the oxidation ratio of nitrogen without the consideration of gas-phase HNO3.

For the discussions of the relationship between SOR and RH, it is important to point out that RH was measured locally at the sampling site but sulfate was mostly formed on a regional scale, ie, in air masses upwind of Beijing. Is it valid to assume that in-situ RH measurement data are representative of the RH conditions in the air masses where sulfate was formed?

The locally measured RH was largely influenced by the RH of the air masses transported from the upwind direction. For example, the RH in Beijing is usually low when the dry northwesterly/northeasterly air masses arrive. When Beijing is influenced by the humid southerly air masses, the RH in Beijing is high (An et al., 2019). Therefore, in-situ RH measurements are representative of the RH conditions in the air masses where sulfate was formed. Relationship between sulfate and the locally measured RH has been discussed a lot by previous studies (Zheng et al., 2015; Li et al., 2017; Fang et al., 2019).

Line 330 -331, does the higher SOR in 2017 than 2014 necessarily demonstrate "a higher sulfate production rate in 2017"? A relatively larger contribution from background air masses could also lead to higher SOR.

With the implementation of the clean air actions,  $SO_2$  emissions were reduced in both Beijing and the surrounding regions. According to back trajectory analysis, Beijing was less influenced by polluted air masses from regional transport in 2017 and sulfate contributed a smaller fraction in the air masses transported from surrounding areas. Therefore, the higher SOR in 2017 than 2014 indicates the higher sulfate production rate in 2017.

Line 544- 545, this citation is incomplete

Updated.

Figures in the supplementary materials are fuzzy, need to use better resolution.

The fuzzy figures in the Supplement have been updated to better resolution.

Figure S6d, explain how to read the figure and the meanings of N-E, W-N, E-S, S-W?

The diurnal pattern in Figure S6d showed the diurnal variations in the contributions of different wind directions. The meanings of N-E, W-N, E-S, and S-W were already explained in the manuscript as "from north to east (N-E;  $0^{\circ} \leq WD < 90^{\circ}$ ), east to south (E-S;  $90^{\circ} \leq WD < 180^{\circ}$ ), south to west (S-W;  $180^{\circ} \leq WD < 270^{\circ}$ ), and west to north (W-N;  $270^{\circ} \leq WD \leq 360^{\circ}$ )".

**Figure S7, what do the color bars stand for?**

The color bars indicate the variations of PM2.5 concentration, which was labelled in the figure.

**References**

Fang, Y., Ye, C., Wang, J., Wu, Y., Hu, M., Lin, W., Xu, F., and Zhu, T.: RH and O3 concentration as two prerequisites for sulfate formation, Atmos. Chem. Phys. Discuss., https://doi.org/10.5194/acp-2019-284, in review, 2019.

Li, H., Zhang, Q., Zhang, Q., Chen, C., Wang, L., Wei, Z., Zhou, S., Parworth, C., Zheng, B., Canonaco, F., Prévôt, A. S. H., Chen, P., Zhang, H., Wallington, T. J., and He, K.: Wintertime aerosol chemistry and haze evolution in an extremely polluted city of the North China Plain: significant contribution from coal and biomass combustion, Atmos. Chem. Phys., 17, 4751-4768, https://doi.org/10.5194/acp-17-4751-2017, 2017.

Li, H., Zhang, Q., Zheng, B., Chen, C., Wu, N., Guo, H., Zhang, Y., Zheng, Y., Li, X., and He, K.: Nitratedriven urban haze pollution during summertime over the North China Plain, Atmos. Chem. Phys., 18, 5293-5306, https://doi.org/10.5194/acp-18-5293-2018, 2018.

Sun, P., Nie, W., Chi, X., Xie, Y., Huang, X., Xu, Z., Qi, X., Xu, Z., Wang, L., Wang, T., Zhang, Q., and Ding, A.: Two years of online measurement of fine particulate nitrate in the western Yangtze River Delta: influences of thermodynamics and N2O5 hydrolysis, Atmos. Chem. Phys., 18, 17177-17190, https://doi.org/10.5194/acp-18-17177-2018, 2018.

Wang, Y., Zhang, Q., Jiang, J., Zhou, W., Wang, B., He, K., Duan, F., Zhang, Q., Philip, S., and Xie, Y.: Enhanced sulfate formation during China's severe winter haze episode in January 2013 missing from current models, Journal of Geophysical Research: Atmospheres, 119, 10,425-410,440, 10.1002/2013JD021426, 2014.

Xu, Q., Wang, S., Jiang, J., Bhattarai, N., Li, X., Chang, X., Qiu, X., Zheng, M., Hua, Y., and Hao, J.: Nitrate dominates the chemical composition of PM2.5 during haze event in Beijing, China, Sci Total Environ, 689, 1293-1303, https://doi.org/10.1016/j.scitotenv.2019.06.294, 2019a.

Xu, W., Sun, Y., Wang, Q., Zhao, J., Wang, J., Ge, X., Xie, C., Zhou, W., Du, W., Li, J., Fu, P., Wang, Z., Worsnop, D. R., and Coe, H.: Changes in Aerosol Chemistry From 2014 to 2016 in Winter in Beijing: Insights From High-Resolution Aerosol Mass Spectrometry, Journal of Geophysical Research: Atmospheres, 124, 1132-1147, 10.1029/2018JD029245, 2019b.

Zheng, G. J., Duan, F. K., Su, H., Ma, Y. L., Cheng, Y., Zheng, B., Zhang, Q., Huang, T., Kimoto, T., Chang, D., Pöschl, U., Cheng, Y. F., and He, K. B.: Exploring the severe winter haze in Beijing: the impact of synoptic weather, regional transport and heterogeneous reactions, Atmos. Chem. Phys., 15, 2969-2983, https://doi.org/10.5194/acp-15-2969-2015, 2015.

**Response to Short Comment 1 (SC1) on "Rapid transition in winter aerosol composition in Beijing from 2014 to 2017: response to clean air actions" by H. Li et al.**

(a) Overview The authors used fine particle data collected in Beijing during the winters of 2014 and 2017 to understand changes in the chemical composition of the PM over a period when particle mass decreased significantly. The results here are consistent with past data, showing a large decrease in PM mass, but also revealing that this is largely because of decreases in sulfate, organics, and unidentified components, as a result of decreasing local and regional emissions. The authors find that meteorology also contributed to lower PM concentrations in 2017 but that this was a smaller factor.

This is an interesting paper with a number of complementary components. On the other hand, there are a number of issues that need to be addressed, as described below. The thermodynamic modeling seems problematic and needs more clarification and information.

We would like to thank Cort Anastasio for giving the constructive and helpful comments and suggestions. In the following, we will answer the comments point by point.

(b) Major Points 1a. Figure 2 and lines 190 - 191: The text states that "All SIA species showed similar diel trends in the two winters: : :". For some species this is true, but for many of the species, the diel variability is much lower in 2017 than in 2014, suggesting that while more local sources dominated in 2014, regional sources are more important in 2017. Take CCOA as an example: in 2014, the concentration varied by a factor of 3, while in 2017 the variation was probably only a factor of 1.5 or less. Can the change in diel variability be used as an indicator of local vs. regional pollution?

SIA species refer to sulfate, nitrate, and ammonium. For the SIA species, they showed similar diel trends in the winters of 2014 and 2017. For many of the other species, especially the primary species, their diel variability was much lower in 2017 than in 2014. However, the changes in the diel variability of these primary species can not be easily used as an indicator of local vs. regional pollution. For example, previous studies reported that coal combustion sources in Beijing were mainly attributed to regional transport from surrounding areas (Wang et al., 2011; Shang et al., 2018). Therefore, we could not rely on the diel variability of CCOA to provide information of local vs. regional pollution.

1b. Of course boundary layer height is a major influence on concentrations as well. I suggest you add a plot of CO as a tracer for BL height. Does plotting the ratio of PM component/CO help disentangle chemistry and BL height? These might be useful supplemental plots.

The ambient variations of PM components are not only controlled by atmospheric chemical reactions but are also influenced by boundary layer developments. According to the comment, we analyzed the CO-scaled concentrations for PM components to eliminate the influence of different dilution/mixing conditions. The corresponding diurnal plots are shown below. By plotting the ratio of PM component/CO, we can more clearly see the photochemical production of sulfate, nitrate, ammonium, and OOA during daytime. These diurnal plots are included in the supplementary information as Fig. S15.

---

## Author Response (AR2)

The authors have done a generally good job responding to review comments and revising the manuscript. However, there are two comments to which the authors' responses do not seem satisfactory.

For the comment that questioning the rationales for using SO4/BC and NO3/BC ratios in the analysis, the authors responded that "The absolute concentration of sulfate and nitrate in the atmosphere is not only controlled by atmospheric chemical reactions but also influenced by boundary layer developments. By using SO4/BC and NO3/BC in the analysis, we can see more clearly how the secondary formation of sulfate and nitrate varied compared to primary emissions." This is assumption is not correct. Boundary layer height affects locally emitted primary pollutants and regionally formed secondary pollutants in different ways. Diel variation in BLH is also associated with changes in air temperature, which may affect semivolatile and nonvolatile species differently. Arguing for sulfate and nitrate formation based on variations in SO4/BC and NO3/BC ratios is misleading. For example, for a secondary species that has a flat diel profile, such as sulfate, its ratio to BC may show a daytime increase only because the dominator (i.e. BC) has a daytime decrease. Related discussions should be removed or revised.

For the comment that the incorrect usage of "nitate/(nitrate+NOx)" as an index for the oxidation ratio of nitrogen. The authors responded "According to the ISORROPIA calculations in this study, the fraction of particulate nitrate in total nitrate was higher than 0.99 for the average winter conditions in both 2014 and 2017. Therefore, it is meaningful to use nitrate/(nitrate+NOx) to evaluate the oxidation ratio of nitrogen without the consideration of gas-phase HNO3." The issue is that oxidation of NOx produces not just HNO3 and nitrate. It also forms organic nitrates. Does the total nitrate from ISORROPIA include gaseous and particulate organic nitrates? This reviewer disagrees it is proper to use nitrate/(nitrate+NOx) in the discussions and request that discussions related to NOR be removed.

Response:

According to the reviewer's suggestions, technical corrections have been made in the revised manuscript. In Sect. 3.2.4, discussions related to the rationales of SO4/BC and NO3/BC have been removed. Discussions related to NOR have also been removed. Figure 8 has been modified accordingly.

[revised manuscript text omitted]